# Adversarial Robust and Explainable Network Intrusion Detection Systems Based on Deep Learning

**Kudzai Sauka, Gun-Yoo Shin, Dong-Wook Kim and Myung-Mook Han ***

School of Computing, Gachon University, Seongnam-si 13120, Korea, 201940084@gachon.ac.kr (K.S.); bobo7754@gachon.ac.kr (G.-Y.S.); kog73006@gachon.ac.kr (D.-W.K.)

* Correspondence: mmhan@gachon.ac.kr

**Abstract:** The ever-evolving cybersecurity environment has given rise to sophisticated adversaries who constantly explore new ways to attack cyberinfrastructure. Recently, the use of deep learning-based intrusion detection systems has been on the rise. This rise is due to deep neural networks (DNN) complexity and efficiency in making anomaly detection activities more accurate. However, the complexity of these models makes them black-box models, as they lack explainability and interpretability. Not only is the DNN perceived as a black-box model, but recent research evidence has also shown that they are vulnerable to adversarial attacks. This paper developed an adversarial robust and explainable network intrusion detection system based on deep learning by applying adversarial training and implementing explainable AI techniques. In our experiments with the NSL-KDD dataset, the PGD adversarial-trained model was a more robust model than DeepFool adversarial-trained and FGSM adversarial-trained models, with a ROC-AUC of 0.87. The FGSM attack did not affect the PGD adversarial-trained model's ROC-AUC, while the DeepFool attack caused a minimal 9.20% reduction in PGD adversarial-trained model's ROC-AUC. PGD attack caused a 15.12% reduction in the DeepFool adversarial-trained model's ROC-AUC and a 12.79% reduction in FGSM trained model's ROC-AUC.

**Keywords:** machine learning; adversarial attacks; explainable; network intrusion detection system; deep neural networks (DNN); adversarial robust

## 1. Introduction

Cyber security has recently faced enormous attention from many researchers in the IT research community. The growth of computer computational power and the wide distribution of the internet of things have put cyber security in the limelight. With the advancement of IT infrastructure and telecommunication facilities distribution, there is a noticeable growth of cyber-attacks continually exploiting the weakness of the cybersecurity infrastructure. Network intrusion detection systems have recently become more popular because of their capabilities for detecting anomaly activities. These architectures are a significant component in the security infrastructure against network attacks. Cyber security researchers have been working tirelessly on new ways to counter the recent attacks with the increased complexity of the attacks. In addition, much research has been performed to augment their detection capabilities by incorporating machine learning and deep learning techniques into IDS.

This incorporation has improved the NIDS detection capabilities even for zero-day attacks. However, new research findings from the mainstream ML and DL have shown that ML and DL models are vulnerable to adversarial attacks, defined as carefully crafted imperceptible changes of inputs used to fool or weaken the capabilities of ML or DL [1]. Adversarial examples can be prepared by intentionally imputing small perturbations to the original inputs. leading to the misclassification of deep learning models with high

confidence. Fast gradient sign method (FGSM) [2], Projected Gradient Descent (PGD) [3], Jacobian-based saliency map attack (JSMA) [4], DeepFool [5], and C&W [6]] attack are representative adversarial examples of generation methods.

Several adversarial defenses and attacks have been crafted and studied in different domains such as malware detection, computer vision, and voice recognition. However, some of these attacks are deemed ill-suited for intrusion detection systems. Hence, there have been few research works on implementing some attacks and defenses from other domains in intrusion detection systems. The main reason given by the IDS research community is that attacks in other fields modify feature vectors instead of real input space. In addition, in crafting attacks for IDS, there is a need to ensure no communication violation or compromise of maliciousness when modifying malicious traffic, which is not a problem for other non-security domains [7]. Regaki et al. [8] studied adversarial examples in NIDS. They showed that adversarial examples generated by FGSM and JSMA methods could significantly reduce the accuracy of deep learning models applied in NIDS. Wang [9] explored the vulnerabilities of deep learning-based intrusion detection systems with JSMA attack, FGSM attack, DeepFool attack, and C&W attack. Their results further proved that the attacks algorithm proposed to fool the deep learning-based image classification can also be employed in intrusion detection.

### 1.1. Anomaly-Based NIDS

Anomaly-based NIDS consists of three major components: traffic capture, feature engineering, and classification, as shown in Figure 1. The feature extractor receives a stream of packets from the monitoring network, extracts features from them, and then feeds them to the anomaly detector. This process also involves an initial step of data preprocessing to make the data ready for the DL-based NIDS for good predictions. Next, the anomaly detector processes the data and finally gives a score for each input it had received, compared to the threshold. The threshold acts as a boundary between malicious and benign inputs; if the score is less than the threshold, it will be deemed benign, and if it is greater than the threshold, the input is malicious [10]. This study focused on crafting an adversarial, robust, and explainable deep learning-based NIDS.

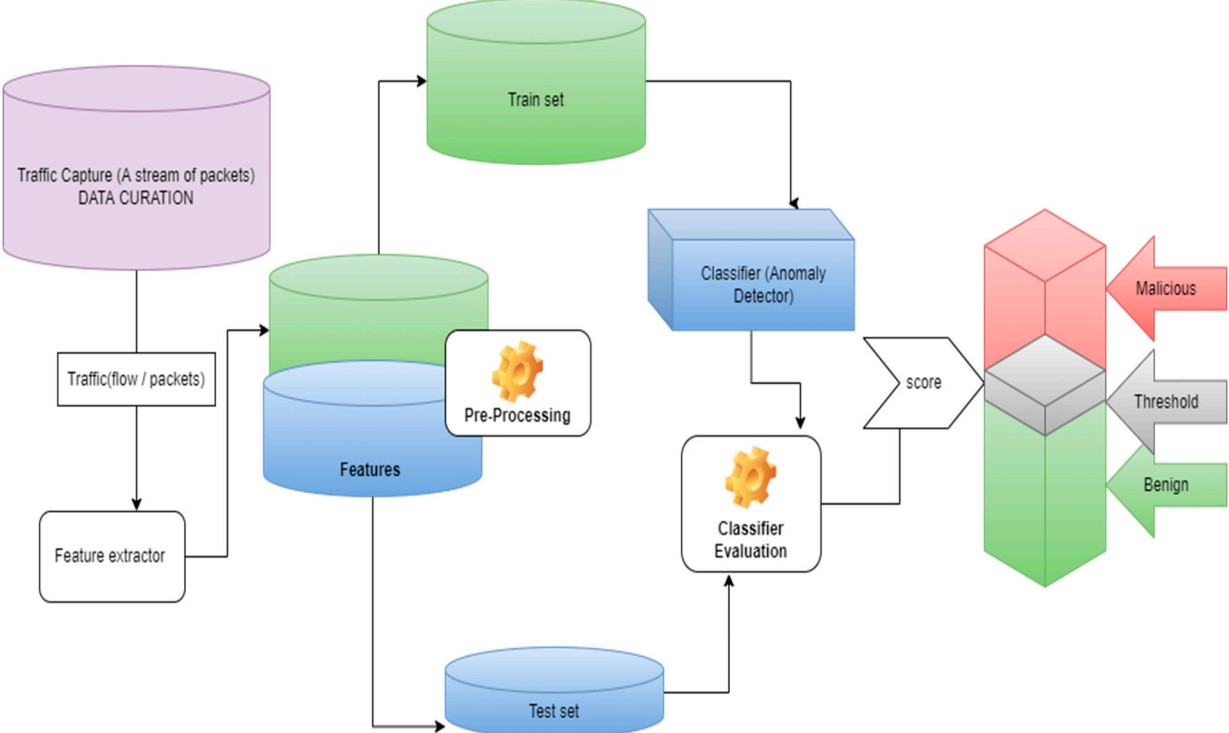

**Figure 1.** Anomaly-based NIDS.

## 1.2. Explainable Deep Learning-Based NIDS

The black-box nature of deep learning models has given rise to trust issues regarding how these models are making predictions, resulting in many organizations being reluctant to implement deep learning-based NIDS [11]. This predicament has sparked research in the field of explainable artificial intelligence (XAI) to aim at making insightful explanations of the internal operation of ML and DL models [12].

The interpretability of AI models can generally be grouped into two major categories: intrinsic interpretability and post hoc interpretability, where intrinsic interpretability is derived by directly incorporating interpretability into the model's structure, which some self-explanatory models such as decision trees, rule-based models, and attention models do. Conversely, the post hoc requires creating a surrogate model to provide explanations from an existing model. Xia Hu [12] et al. further differentiated the AI interpretability into global interpretability and local interpretability, where local interpretability refers to the local explanation of an individual prediction of a model, revealing how some decisions are derived. AI interpretability will aid in unveiling the relationship between specific input and its matching output. Global interpretability involves the inspection of the structure and parameters of a complex model to understand the modalities of the model.

Global interpretability demystifies the internal mechanism of AI models, thereby increasing their transparency. Given this growing trend of XAI and its applicability in other domains such as computer vision, it is also imperative to apply these technologies into deep learning-based NIDS to demystify them, thus encouraging operational deployment. There are many recommended ways to generate model explanations. This paper used the SHapley Additive exPlanations (SHAP) [8] method. SHAP combines local and global interpretability simultaneously, thus improving the interpretation of IDSs.

## 1.3. Adversarial Machine Learning

Szegedy et al., 2013 [1], and Goodfellow et al., 2014 [2], pioneered the research of DNN vulnerability, and they indicated that adversarial samples could easily fool DNNs. An adversary formulates adversarial examples by applying almost imperceptible human perturbations to examples from the dataset. These perturbations lead the model to make wrong predictions with high confidence. Adversarial machine learning is crafting perturbations (*adversarial samples)* to fool machine learning models. Adversarial sample generation can be performed at all phases of machine learning models, as shown in Figure 2. An attacker can craft perturbations by modifying input data during training or prediction phases.

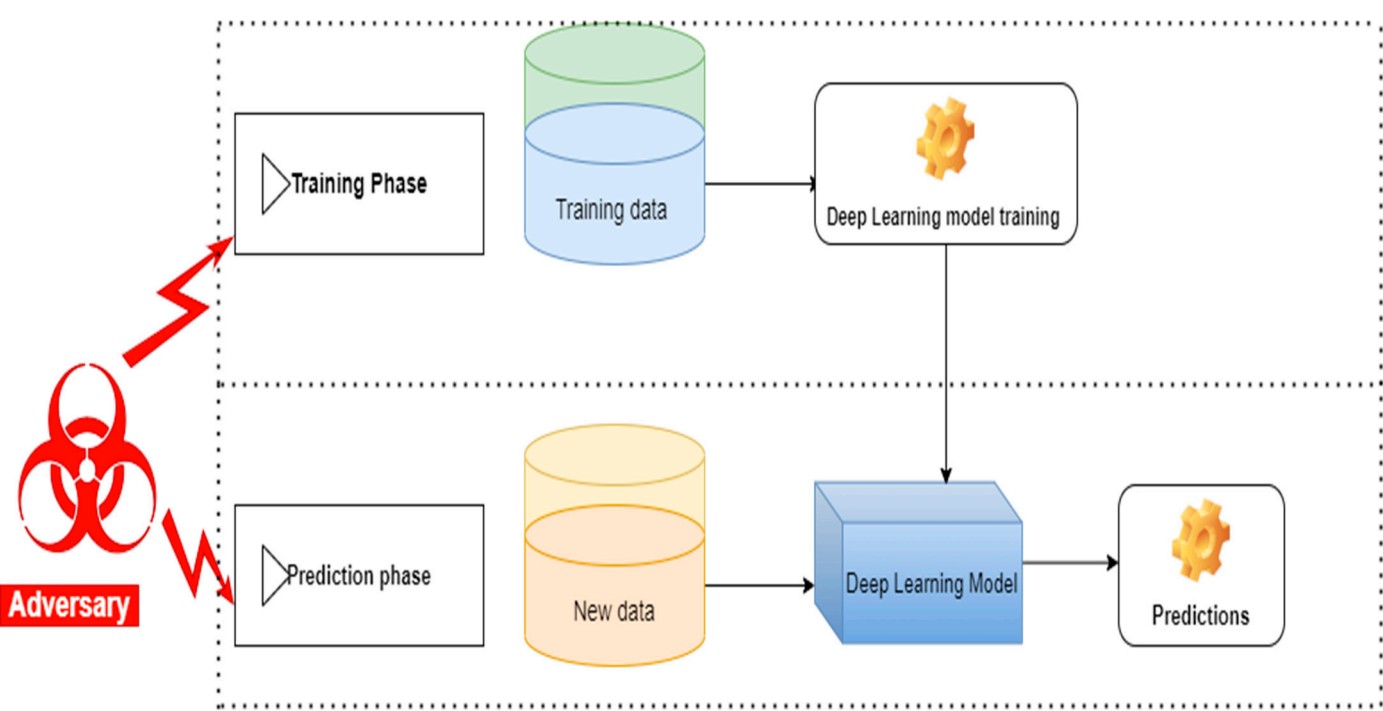

**Figure 2.** Adversarial machine learning. Reproduced with permission from [13], CEUR (http://ceur-ws.org/Vol-2057/), 2017.

Adversarial attacks can be categorized based on the adversary's goal or based on the adversary's knowledge. Classification of adversarial attacks based on adversarial goals can be further classified into poisoning attacks and evasion attacks [14], where poisoning attacks refer to a process where an adversary modifies the training dataset by inputting malicious samples. During an evasion attack, adversaries do not have the power or access to change the model or its parameters, but they can create malicious samples foreign to the model; hence during testing, the classifier will not recognize these samples resulting in wrong classifications. In addition, an attack can be targeted or non-targeted under the adversary's goal classification. A targeted attack is where the adversary's objective is to trick the model into producing the specified output. While a non-targeted attack, the adversary's goal is to make the model perform poorly by misclassifying some samples with high accuracy.

Classification based on the adversary's knowledge can also be further categorized into white-box attacks, black-box attacks, and grey-box attacks based on the amount of information the adversary has on the model. A white-box attack refers to a situation where the adversary has full access to the target model. They have information about the model at their disposal. Under the black-box attack scenario, the adversary does not have access to the model's inner configuration. They can only input data and query the output of the model, and then the adversary can be able to create their dataset, which can be used to develop surrogate models. Finally, they can use transfer learning techniques to develop adversarial samples for a target model. In the grey-box scenario, the adversary has partial knowledge about the model architecture but does not have access to the weights in the model [15]. This paper used evasion, non-targeted and white-box attacks for the proposed framework evaluation. The attacks are FGSM, PGD, and DeepFool.

### 1.3.1. Fast Gradient Sign Method (FGSM)

FGSM was proposed by Goodfellow et al. [2] with an obligation of generating a perturbation $\eta$ by computing the gradient of the cost function $J$ with respect to the input $x$. Instead of a leaner search to find the optimal value of the perturbation, Goodfellow et al. [2] proposed a one-step gradient update along the direction of the gradient sign. This can

be performed efficiently using backpropagation. Their perturbation can be expressed as (1) where $\theta$ represents the model's parameters, $\epsilon$ is the perturbation size, $x$ represents the model inputs, and $y$ represents targets associated with $x$ [2] performed some experiments on GoogLeNet. They managed to fool GoogLeNet into classifying a Panda as a Gibbon by adding an imperceptibly small vector whose elements are equal to the sign of the elements of the gradient of the cost function with respect to the input, as shown in Figure 3.

$$\eta = \epsilon \, sign(\nabla_x J_\theta(x, y)) \tag{1}$$

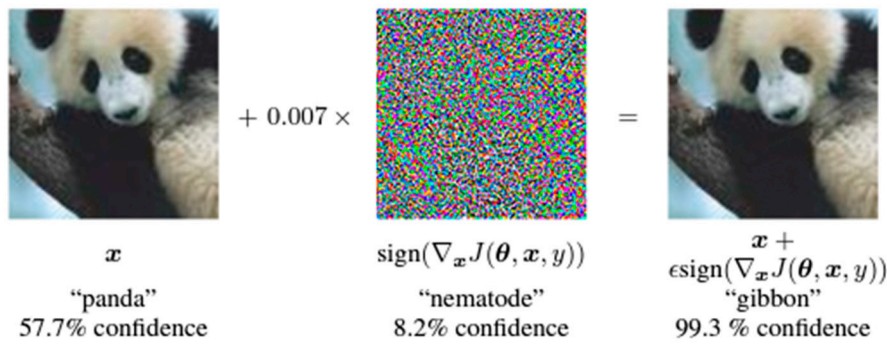

**Figure 3.** An illustration of applying FGSM on GoogLeNet. Adapted with permission from [2], arXiv, 2013

### 1.3.2. Projected Gradient Descent (PGD)

The PGD can be considered as a generalized version of BIM without the constraint $\alpha T = \in$. To constrain the adversarial perturbations, the PGD projects the adversarial samples learned from each iteration in the $\in -L_\infty$ neighbor of the benign samples. Hence, the adversarial perturbation size is smaller than $\in$.

$$x'_{t+1} = Proj\{x'_t + \alpha . sign[\nabla_x J(\theta, x', y)]\} \tag{2}$$

where *Proj* projects the updated adversarial sample into the $\in -L_\infty$ neighbor and a valid range.

### 1.3.3. DeepFool

The DeepFool algorithm was designed to find an adversarial example to an image by finding the closest decision boundary and orthogonally projecting it onto the boundary [5]. Once it crosses the boundary, it will be an adversarial image, as shown in Figure 4, where there is $f$, an affine classifier that can be generalized to any differentiable binary classifier (3). It can be easily be seen that the robustness of point (4), is equal to the distance from $x_0$, the separating affine hyperplane (5). The minimal perturbation to change the classifier's decision corresponds to the orthogonal projection of $x_0$ onto $F$. In the case of $f$ being a general binary differentiable classifier, an iterative procedure to estimate the robustness $\Delta(x_0; f)$ is adopted.

$$f(x) = w^T x + b \tag{3}$$

$$x_0, \Delta(x_0; f)^2 \tag{4}$$

$$F = \{x : w^T x + b = 0\} \tag{5}$$

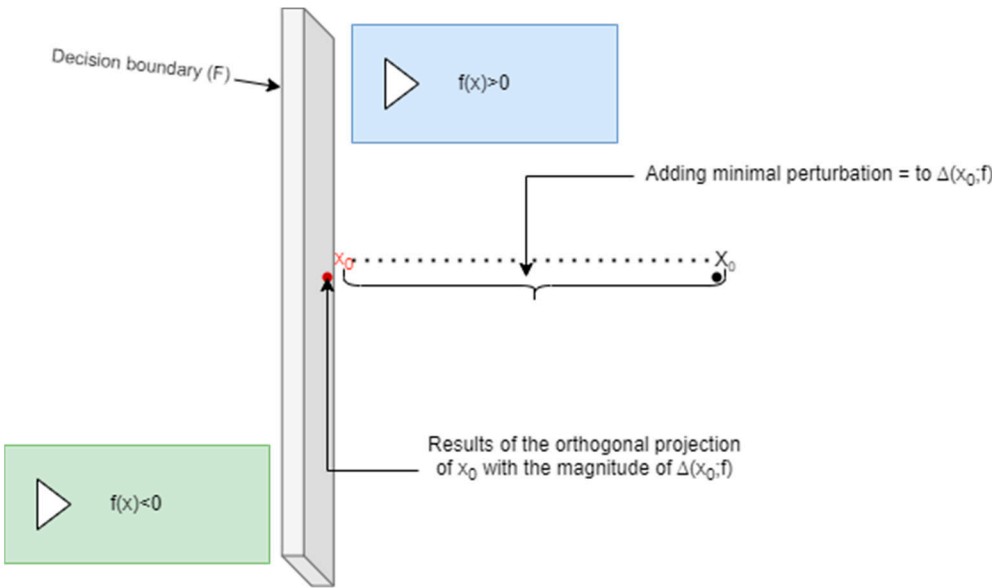

**Figure 4.** DeepFool adversarial examples for a linear binary classifier illustration.

### 1.3.4. Carlini and Wagner attack (C&W)

Carlini and Wagner proposed a set of optimization-based adversarial attacks (C&W attacks) that can generate $L_0, L_2, L_\infty$ norm-measured adversarial samples, namely $CW_0, CW_2, CW_\infty$.

The optimization objective is as follows:

$$\min_\delta D(x, x + \delta) + c \cdot f(x + \delta) \; subject \; to \; x + \delta \in [0,1] \tag{6}$$

where: $\delta$ denotes the adversarial perturbation, $D(\cdot, \cdot)$ denotes the distance metric $L_0, L_2, L_\infty$, $f(x + \delta)$ denotes a customized adversarial loss that satisfies $f(x + \delta) \leq 0$ if the DNN's prediction is the attack target.

### 1.3.5. Jacobian-Based Saliency Map Approach (JSMA)

Papernot et al. proposed an efficient target attack that can fool DNNs with small $L_0$ perturbations. The method first computes the Jacobian matrix of the logit outputs $l(x)$ before the SoftMax layer.

$$\nabla l(x) = \frac{\partial l(x)}{\partial x_y} = \left[ \frac{\partial l_j(x)}{\partial x_\gamma} \right]_{\gamma \epsilon 1 \ldots M_{in} \; j \epsilon 1 \ldots M_{out}} \tag{7}$$

where $M_{in}$ is the number of neurons on the input layer; $M_{out}$ is the number of neurons on the output layer; $\gamma$ is the index for input $x$ component; $j$ is the index for output $l$ component.

### *1.4. Adversarial Robustness*

Adversarial robustness entails efforts to defend neural networks against adversarial inputs. Defense against adversarial attacks can be grouped into three categories: gradient masking, robust optimization, and adversarial examples detection [14]. Gradient masking is when the model's gradient information is deliberately hidden to confuse the adversaries, as most attack algorithms depend on the model's gradient information [14]. Methods for gradient masking include: defensive distillation [16], shattered gradient [17], randomized gradients [18]. Carlin and Wanger [6] performed experiments against defensive distillation; their results revealed that defensive distillation is effective against the Deep-Fool, Fast gradient method, and JSMA-F attack but not against C&W attack ($l_0, l_2, and \; l_\infty$). Xu et al. [14] indicated that the main weakness of gradient masking is that it can only confront the adversaries, but it cannot eliminate their existence.

Adversarial example detection involves the study of normal samples distribution and then using the knowledge to detect adversarial examples and disallowing their inputs into the model.

Robust optimization involves modifying the way the neural network model learns its parameters. Adversarial training is the primary technique under robust optimization, which [1,2] indicates that it is the most effective defense against adversarial attacks. Adversarial training injects adversarial examples into the training set to make it more robust to attack or to reduce its test error on clean inputs [1]. Most adversarial training studies were evaluated using the same adversarial attack, which would have been used to generate the adversarial example. Goodfellow et al. [2] tested an FGSM-trained model on an FGSM-generated adversarial sample. Mnady et al. propose adversarial training using a PGD attack. Their model was resistant to FGSM, PGD, and C&W. Debicha et al. [19] used PGD to examine the effectiveness of adversarial training, making the intrusion detection systems robust against adversarial attacks. This paper proposes a different approach to developing an adversarial robust deep learning-based network intrusion detection system by retraining our model with DeepFool adversarial samples and then testing the model against FGSM and PGD methods.

Although several researchers in the cybersecurity community have been focusing on the detection accuracy of various ML-based NIDS, a few have focused on the explainability and interpretability of deep learning-based intrusion detection systems and how they can be made robust against adversarial attacks.

This research work used the NSL-KDD dataset to develop a robust and explainable NIDS based on deep learning. We used three untargeted and white-box attacks to generate adversarial examples for the experiment. We demonstrated how adversarial attacks could undermine classical multi-class machine learning-based NIDS as well as deep learning-based NIDS, and this confirms the work of [7–10,20]. Hence, we find justification for creating an adversarial robust deep learning model that is less affected by adversarial attacks. We proposed a measure of adversarial robustness, Equation (12), for robustness comparison. We implemented the SHAP technique to explain robust adversarial NIDS based on DL to extract important features used by the model to make classification decisions. We strongly believe that this is the first paper to implement a combination of explainable AI techniques and adversarial learning into NIDS.

The rest of the paper is structured as follows: The methods and materials used are outlined in Section 2. The experiment and its results are presented in Section 3. Section 4 presents the discussion of the research results. Finally, the conclusion and suggestions for future works are presented in Section 5.

## 2. Materials and Methods

*Proposed Method*

This paper builds deep learning-based NIDS to create a state-of-the-art adversarial robust and explainable deep learning-based NIDS. The focus was on white-box untargeted attacks: FGSM, PGD, and DeepFool attacks. Throughout the adversarial learning literature [14], it has been outlined that an efficient way to make deep learning models robust against adversarial attacks is to train them with a mixture of adversarial samples and clean training data. This process is known as adversarial training, as shown in Figure 5.

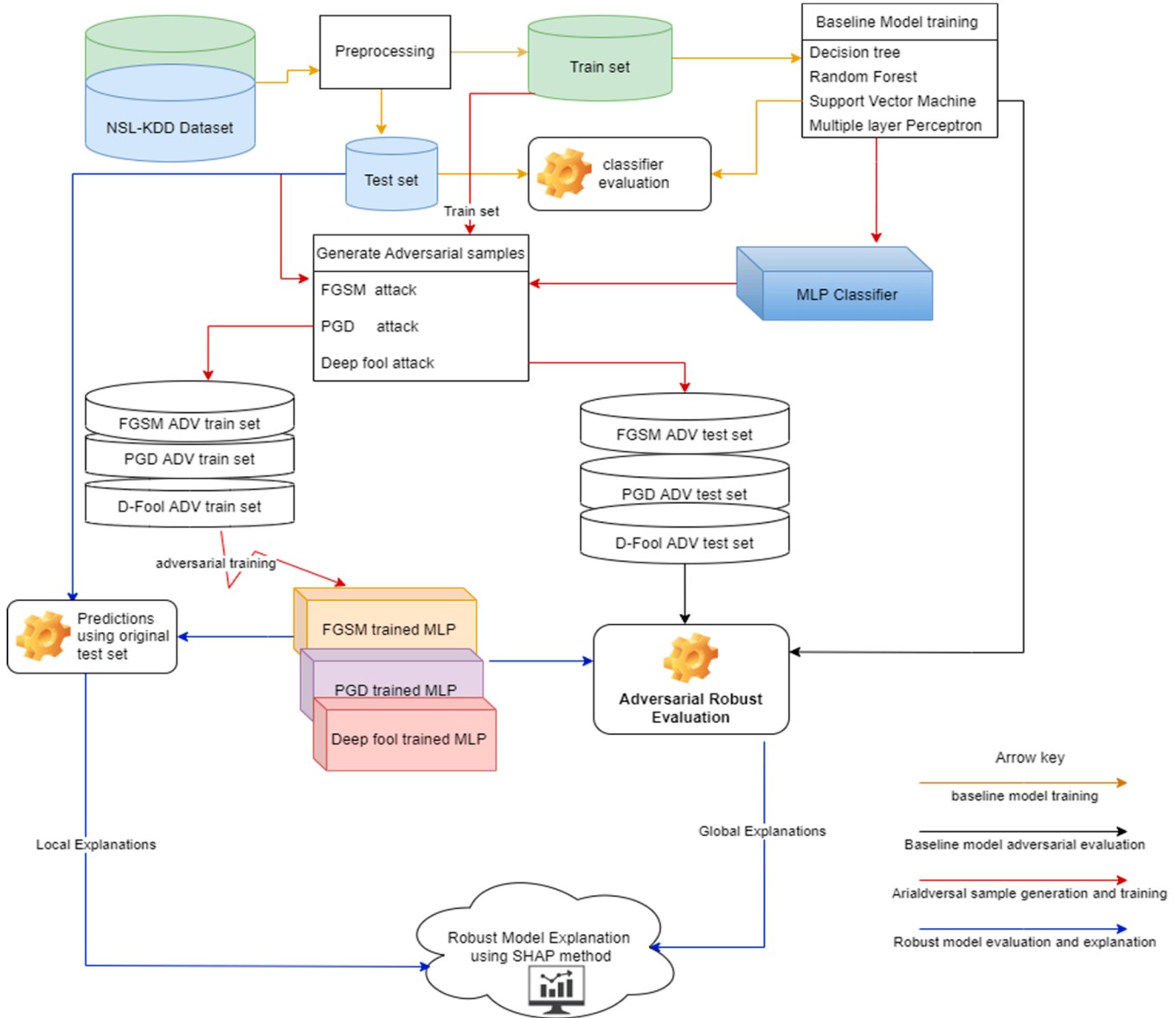

**Figure 5.** Proposed framework for adversarial robust and explainable DL-based NIDS.

The first stage of the proposed framework is to collect the network security dataset that contains different attacks. NSL-KDD dataset was chosen for the purposed of carrying out the experiments. After collecting the dataset, it is then preprocessed, where feature modification is performed and redundancy is lowered. At this stage, separation of train and test set is then performed to set aside the dataset for model evaluation.

After preprocessing, baseline model training then starts. In this study, we built four baseline models, three classical machine learning multi-class classifiers, and a multi-layer perceptron mode, representing a deep learning model. After model training, model evaluation is performed using AUC.

The third stage is the generation of adversarial samples. Again, the baseline multi-layer perception model generates adversarial examples using different types of adversarial attacks. This study generated three different adversarial samples from three different adversarial attacks: FGSM, DeepFool, and PGD. The adversarial samples include pairs of the adversarial test samples and adversarial train set, which were generated using benign test seta and train set, respectively. An adversarial train set will be used for adversarial training, while an adversarial test set will be used for a robust adversarial test.

Following adversarial sample generation, the next stage will be to evaluate the adversarial robustness of baseline models using our proposed adversarial robust evaluation measure.

After baseline model adversarial robust evaluation, the next stage is the adversarial training of the baseline multi-layer perceptron using the available adversarial attacks. We developed three distinct adversarial-trained models in this case: FGSM adversarial-trained model, DeepFool adversarial-trained model, and PGD adversarial-trained model.

After adversarial training, we apply our proposed adversarial robust evaluation measure to check which of the adversarial-trained model is more robust. The final stage is a robust model explanation using the SHAP method.

All our models were implemented using TensorFlow 2.8.0 and Keras. The experiments were performed on a Virtual machine with GPU acceleration with 32 GB memory and three Intel core processors at 3.00 GHz. For generating adversarial samples, we used the open-source IBM Robustness Toolbox (ART) framework [21].

## 3. Experiments and Results

In this section, we first examined the detection capabilities of the baseline model; Decision Tree Classifier, Random Forest Classifier, Linear SVM Classifier, and the baseline MLP model under normal samples (without adversarial examples). These baseline models serve as experiment controls. Then, we examined the adversarial sample generation capabilities of the adversarial attacks under review. Third, we evaluated the applicability of DeepFool, FGSM, and PGD adversarial attacks on baseline models. After adversarial sample generation, we retrained our baseline MLP model with an aggregated training set: a mixture of benign and adversarial samples. The process resulted in three new distinct MLP models: DeepFool-trained model, FGSM-trained model, and PGSM-trained model. After MPL adversarial training, the adversarial robustness stage followed; we used Equation (12) to evaluate adversarial robustness. The more robust model against all other attacks was selected as an adversarial robust model. Finally, we applied the SHAP XAI technique to explain the predictions of adversarial robust NIDS based on the deep learning model.

### 3.1. IDS Datasets Selection

The performance of ML/DL-based NIDS has been evaluated on several security datasets, although, out of the IDS research fraternity, these datasets include KDD-Cup'99, NSL-KDD, UNSW-N15, DARPA, DEFCON, CDX, Kyto, etc. [22]. This study was performed using the NSL-KDD dataset, an updated version of the KDD Cup'99 dataset, with many replicated records. NSL-KDD was selected because its train and test set the number of records to be logical, making it comfortable to perform the experiments on the entire dataset without randomly splitting it into small segments. In addition, previous researchers have widely used it, and it is more robust than previous versions of the same data (KDD'99).

The features in the NSL-KDD dataset have three data types: nominal, binary, and numeric. *Binary data* can be viewed as variables that contain numeric values since a numeric value is enough to indicate the presence (1) or absence (0) of a specific status. *Nominal data* are variables that contain categorical values rather than numeric values. The NSL-KDD has 148,515 records divided into a training set with 125,972 records and a testing set with 22,543 records [9]. The dataset has 41 features. A clear description of these features is outlined in Table 1. The features are in three categories: basic features, traffic features, and content features [9]. Basic features (feature numbers 1 to 9 in Table 1) are related to connectivity information, such as protocols. Feature numbers 10 to 22 are content features, which are features within a connection suggested by domain knowledge. Finally, traffic features (feature numbers 23 to 41) are calculated as an aggregate during a window interval [9].

**Table 1.** Total features in the NSL-KDD dataset. Data from [23,24].

| Feature Number | Feature | Type | Description |
|---|---|---|---|
| 1 | Duration | Numeric | Duration of the connection |
| 2 | Protocol_type | Nominal | Type of the protocol |
| 3 | Service | Nominal | Network service on the destination |
| 4 | Flag | Nominal | Normal or error status of the connection |
| 5 | Src_bytes | Numeric | Number of bytes transferred from source to destination |
| 6 | Dst_bytes | Numeric | number of bytes transferred from destination to source |
| 7 | Land | Binary | 1 if the connection is from/to the same host/port; 0 otherwise |
| 8 | Wrong_fragment | Numeric | number of "wrong" fragments |
| 9 | Urgent | Numeric | number of urgent packets (with the urgent bit set) |
| 10 | Hot | Numeric | number of "hot" indicators |
| 11 | Num_failed_logins | Numeric | number of failed login attempts |
| 12 | Logged_in | Binary | 1 if successfully logged in; 0 otherwise |
| 13 | Num_compromissed | Numeric | number of "compromised" conditions |
| 14 | Root_shell | Binary | 1 if root shell is obtained; 0 otherwise |
| 15 | Su_attempted | Binary | 1 if "su root" command attempted; 0 otherwise |
| 16 | Num_root | Numeric | number of "root" accesses |
| 17 | Num file cre ations | Numeric | number of file creation operations |
| 18 | Num_shells | Binary | number of shell prompts |
| 19 | Num_access_files | Numeric | number of operations on access control files |
| 20 | Num_outbound_cmds | Numeric | number of outbound commands in an ftp session |
| 21 | Is_hot_login | Binary | 1 if the login belongs to the "hot" list: 0 otherwise |
| 22 | Is_guest_login | Binary | 1 if the login is a "guest" login; 0 otherwise |
| 23 | Count | Numeric | number of connections to the same host as the current connection (Note: |
| 24 | Serror_rate | Numeric | number of connections that have "SYN" errors |
| 25 | Rerror_rate | Numeric | % of connections that have "REJ" errors |
| 26 | Same_srv_rate | Numeric | % of connections to the same service |
| 27 | Diff_srv_rate | Numeric | % of connections to different services |
| 28 | Srv_count | Numeric | % of connections to the same service as the current connection in |
| 29 | Srv_serror_rate | Numeric | % of connections that have "SYN" errors |
| 30 | Srv_rerror_rate | Numeric | % of connections that have "REJ" errors |
| 31 | Srv_diff_host_rate | Numeric | % of connections to different hosts |
| 32 | Dst host_count | Numeric | number of connections having the same destination host |
| 33 | Dst_host_srv_count | Numeric | number of connections using the same service |
| 34 | Dst_host_same_srv_ | Numeric | % of connections using the same service |
| 35 | Dst_host_srv_diff_ | Numeric | % of different services on the current host |
| 36 | Dst_host_same_src_ | Numeric | % of connections to the current host having the same src port |
| 37 | Dst_host_srv_diff_ | Numeric | % of connections to the same service coming from different hosts |
| 38 | Dst_host_serror_rate | Numeric | % of connections to the current host that have a so error |
| 39 | Dst_host_srv_serror_rate | Numeric | % of connections to the current host and specified service that |

| 40 | Dst_host_rerror_rate | Numeric | % of connections to the current host that have an RST error |
| 41 | Dst_host_srv_rerror_rate | Numeric | % of connections to the current host and specified service that |

There are four classes in the dataset from 39 different attacks, the test set has a total of 37 attacks, and the train set has a total of 22 attacks, as shown in Tables 2 and 3, respectively: denial of service (DoS), probe, remote to local (R2L and user to Root (U2R)) as presented in Tables 2 and 3.

**Table 2.** Test set attack classification. Data from [23,24].

| Attack Label | Attack Type |
| --- | --- |
| Denial of service (DOS) | Back, Land, Naptune, Pod, Smurf, Teardrop, Apache2, Udpstorm, Processable, Worm Mailbomb |
| Prob | ipsweep ,saint, mscan, satan, nmap, portsweep |
| Remote to local (R2L) | Guess_Password, Ftp_write, Imap, Phf, Multihop,Warezmaster, Xlock, Xsnoop, Snmpguess, Snmpgetattack, Httptunnel, Sendmail, Named |
| User To Root (U2R) | Buffer_overflow, Rootkit, Perl, Sqlattack, Xterm, Ps, loadmodule |

**Table 3.** Train set attack classification. Data from [23,24].

| Attack Label | Attack Type |
| --- | --- |
| Denial of service (DOS) | Back, Land, Naptune, Pod, Smurf, Teardrop, |
| Prob | Buffer_overflow, ipsweep, portsweep, nmap, satan |
| Remote to local (R2L) | Guess_Password, Ftp_write, Imap, Phf, Multihop, Warezmaster, Warezclient, Spy, |
| User To Root (U2R) | Loadmodule, Rootkit, Perl, |

### 3.2. Data Preprocessing

Data preprocessing is one of the essential stages in machine learning; deep learning models' efficiency is highly affected by the general scale of the dataset. It is the work of data preprocessing that ensures that the data are suitable for data modeling. The problem was transformed to a five-class classification by changing the attack from 39 different attacks to four categories, as presented in Tables 2 and 3, and a normal class. These attacks were also one-hot encoded for a multi-layer perceptron model and label encoded for classical machine learning models.

One-hot encoding was also performed to convert all categorical variables to numeric. *Protocol _type* had three distinct categories through one-hot encoding; they were transformed into three new features. *Service* had 70 distinct categories; these were transformed into 70 new features. Finally, *the flag* had 11 categories; these were transformed into 11 new features. After preprocessing the 41 features, the NSL-KDD dataset was transformed into a new dataset with 122 numeric features.

After one-hot encoding, Min-Max scaling was implemented on the dataset such that the features had the same scale, with values ranging from 0 to 1.

### 3.3. Classical Machine Learning Multi-Class Classifiers

This paper aimed to develop a multi-class network intrusion detection system based on deep learning, which is not the case with [19,25–28], whose focus was on binary classification. For baseline models, three classical machine learning multi-class classifiers were chosen based on their popularity in the literature. In addition, the OneVsRest classifier was implemented on the Decision Tree classifier, Support Vector Machine classifier, and Random Forest classifier to make them suitable for a multi-class classification task.

The hyperparameters chosen for Decision Tree classifier were: criterion: 'gini', random_state = 42, and max_depth = 12. For Random Forest, the hyperparameters were: max_depth = 70, max_featuires = 'auto', mini_sample_leaf = 4, min_sample_split = 10 and n-estimators = 400. For Support Vector Machin: C = 1, random_state = 42, loss = 'hinge' were the hyperparameters.

### 3.4. Deep Learning-Based NID

To detect intrusions, we used a Multi-Layer Perceptron (MLP) resembling a deep neural network with three hidden layers and 270 hidden units (120-90-60); we trained and tested it based on the NSL-KDD training and test sets in TensorFlow 2.8.0. The model structure is shown in Figure 6. The input layer is depicted with neurons in green color from $x_1 - x_{122}$. The blue color depicts hidden layers, and *m* represents the total number of neurons per hidden layer (*120 neurons in the first hidden layer, 90 in the second hidden layer, and 60 in the third hidden layer*). The output layer depicted in red color has five neurons, representing the number of our classes. We adopted the information bottleneck principle [29] to ensure a robust classification MLP model; we adopted this principle to extract relevant information from the input features about classes. This approach has also been adopted by [13,19,27,30]. Rectified linear unit (ReLU) was used as an activation function within each hidden unit to introduce non-linearity in these neurons' output. Following each hidden layer, a dropout layer with a dropout rate of 0.5 was employed to prevent neural networks from overfitting. Finally, the SoftMax activation function was added to the output layer to normalize the probability distribution.While compiling the model, categorical _cross entropy was used as a loss function and ADAM as an optimization algorithm.

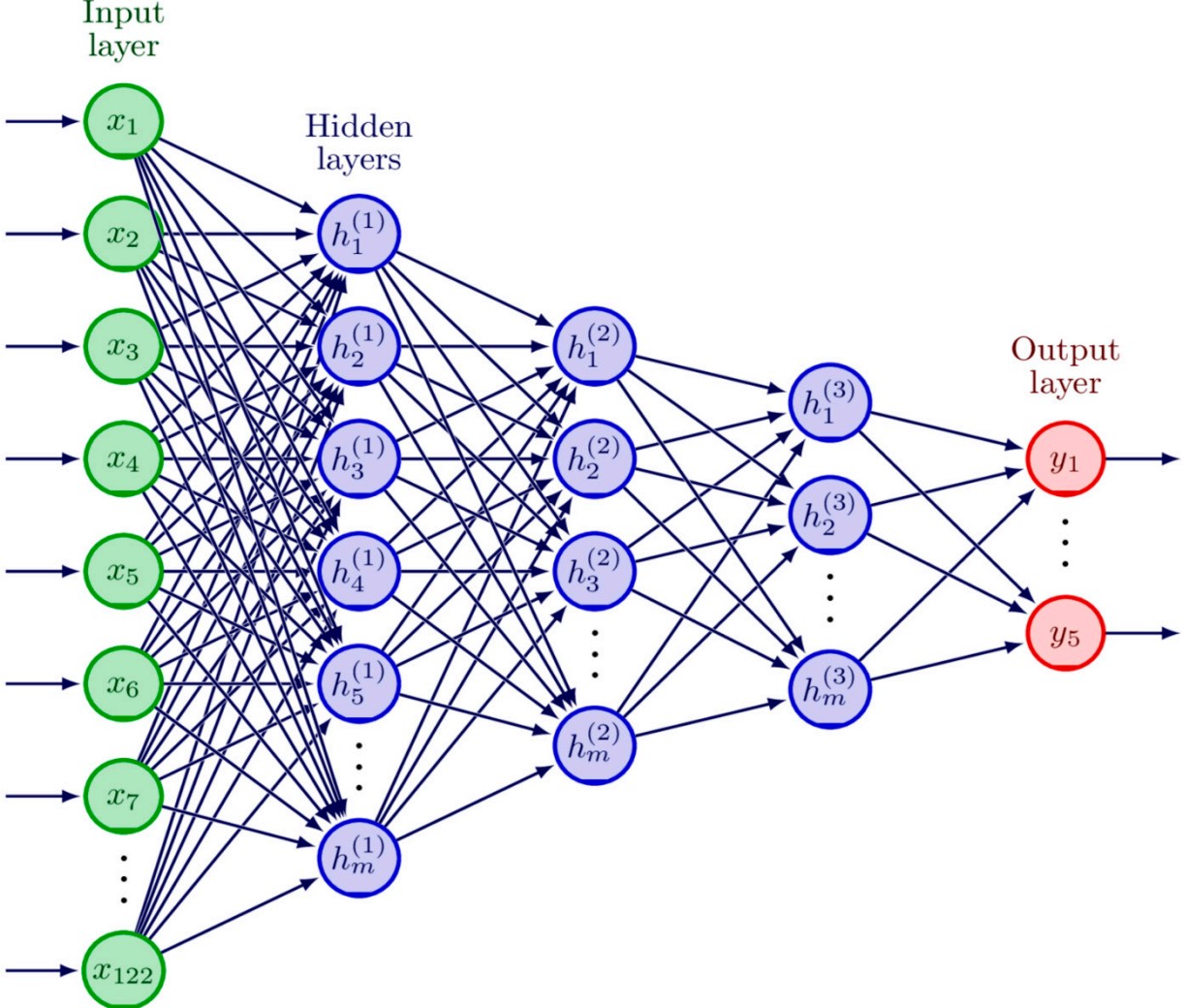

**Figure 6.** Model architecture.

*3.5. Evaluation Metrics*

To evaluate the performance of our baseline models, together with the adversarial-trained models, we applied a different set of evaluation metrics: prediction accuracy, precision, recall, F1 score, and AUC (area under the ROC curve). All these metrics are based on the elements of the confusion matrix: True Positive (*TP*), True Negative (*TN*), False Positive (*FP*), and False Negative (*FN*).

3.5.1. Precision

These metrics indicate the percentage of accurately classified attack samples over all samples classified as attacks. It can be evaluated mathematically as (6) [31].

$$Precision = \frac{TP}{TP + FP} \tag{8}$$

3.5.2. Recall

Recall that (7) represents the percentage of accurately classified attack samples over the total of attack records [31].

$$Recall(True\ Positive\ Rate) = \frac{TP}{TP + FN} \tag{9}$$

### 3.5.3. F1 Score

It is the simultaneous measurement of precision and recall. It uses the harmonic mean technique to calculate the average of precision and recall [31].

$$F1 - Score \ = \ 2 \cdot \frac{Precision \cdot Recall}{Precision \ + \ Recall} \tag{10}$$

### 3.5.4. ROC Curve

It is a graphical presentation of the classification model performance at all classification thresholds. The curve plots *True Positive Rate* against *False Positive Rate* at different thresholds [32]. Where the true positive rate is also called recall, the false-positive rate can be defined as (9).

$$False \ Positive \ Rate \ = \ \frac{FP}{FP \ + \ TN} \tag{11}$$

### 3.5.5. ROC-AUC: Area under the ROC Curve

It is a measure of the area underneath the ROC curve. It provides a combined performance measure across possible classification thresholds [32]. ROC-AUC was our primary evaluation metric because it utilizes probabilities of class prediction, thus helping us to evaluate and compare the models ideally [33]. Our focus was on ROC-AUC as a performance measure of multi-class classification models rather than F1 score and accuracy because our dataset is unbalanced; hence, accuracy was deemed not the best evaluation metric. Hyperparameter: average 'micro' was used to calculate the total ROC-AUC for five classes [34]. Our basis for using AUC as a preferable classifier evaluation measure follows the research of [33,35], who indicated that AUC is a better classifier evaluation measure for either balanced or unbalanced datasets.

### *3.6. Generating Adversarial Samples*

The idea is to use Nicolae et al. (2018) [21] Adversarial Robustness Toolbox (ART) to implement adversarial attacks as well adversarial training. In this case, the adversarial attack can be considered the inverse gradient descent process.

This paper uses the multi-layer perception model in Figure 6 to generate adversarial samples using FGSM, PGD, and DeepFool. The selection of adversarial attacks was based on the availability of their updated version on the ART library, their usability in the network security dataset, and our assumed angle of the adversary. We assumed that the adversary knew the underlying deep learning model's mechanism and did not have specifically targeted output values. Hence, we focused on white-box attacks as well as untargeted attacks. The baseline MLP, FGSM-trained MLP, PGD-trained MLP, and DeepFool-trained MLP were all targets of the adversarial attacks. The three adversarial-trained models and baseline MLP were used as controls during the experiments. While FGSM-trained MLP, PGD-trained MLP, and DeepFool-trained MLP were the models under evaluation.

### Adversarial Robustness

In this paper, we propose a different approach to developing an adversarial robust deep learning-based network intrusion detection system by retraining our model with FGSM, PGD, and DeepFool adversarial samples and then testing the models against the FGSM, PGD, and DeepFool attacks. The adversarial training and test set for adversarial training were generated through heuristic data augmentations [36,37], where the benign trainset was combined with the generated adversarial trainset. In contrast, the label trainset for adversarial training was an augmentation of the original data trainset labels; the illustration is in Figure 7. The idea is to find the most resistant model against the attacks under review. The model that is more resistant to all the attacks is the solution to the adversarial robustness of the network intrusion detection system based on deep learning.

Percentage change of the adversarial-trained model's ROC-AUC and adversarial-attacked model's ROC-AUC was used for adversarial robust evaluation (12). The smaller the negative value, the more robust the model is.

$$Adversarial\ Robust\ Evaluation\ =\frac{AUC_{ij}-AUC_j}{AUC_j}\times\frac{100}{1} \qquad (12)$$

where:

$AUC_{ij}$ is the area under the roc curve of adversarial-trained model $j$ under adversarial attack $i$.

$AUC_j$ is the area under the roc curve of adversarial-trained model $j$.

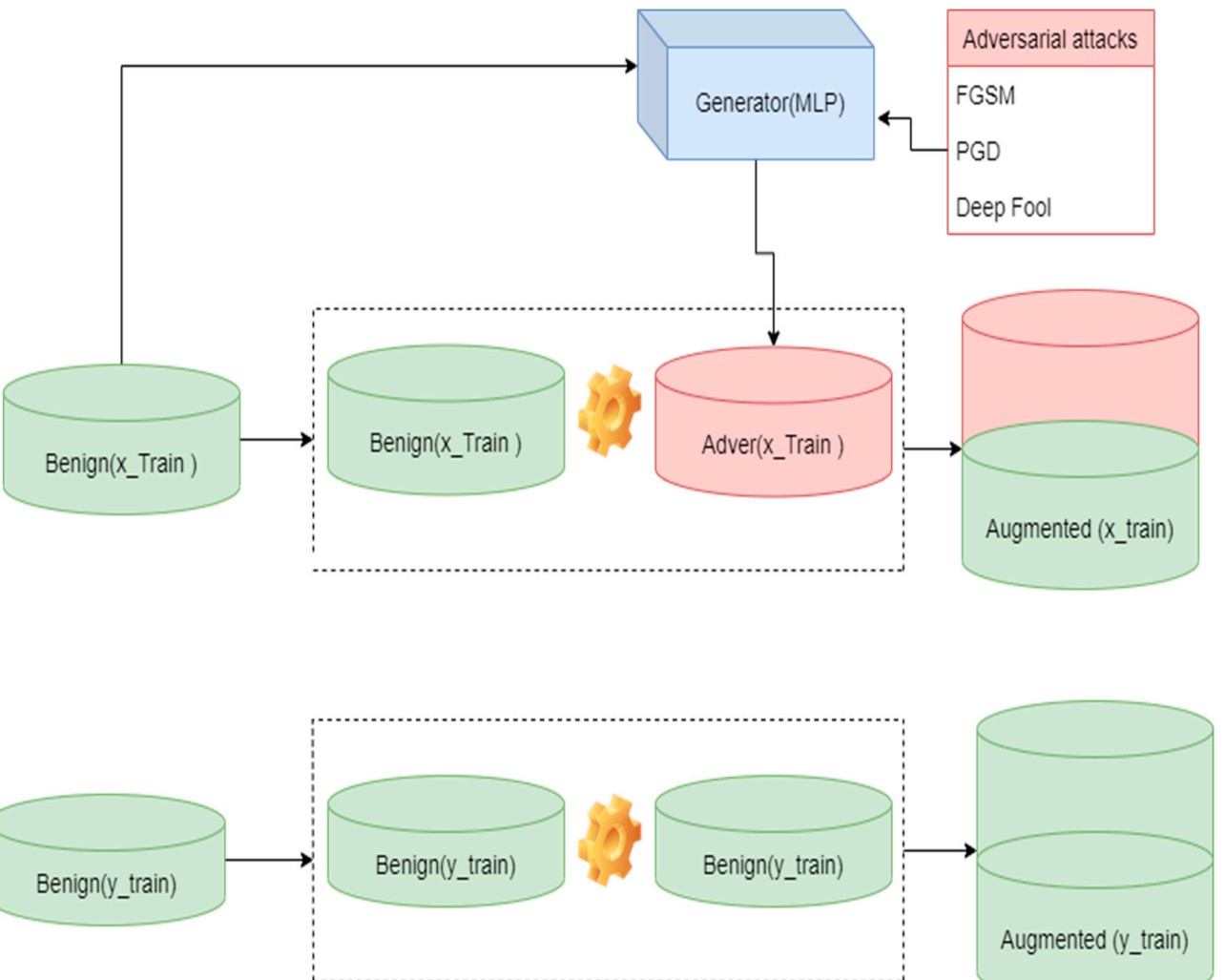

**Figure 7.** Heuristic data augmentation.

### 3.7. Baseline Models Performance Evaluation

Table 4 presents the results of four baseline models used in this research as controls; these include three classical machine learning multi-class classifiers and one baseline MLP model. However, we followed [13] in building classical machine learning multi-class classifiers and built our specific multi-layer perceptron model. The average ROC-AUC for all the models was 0.81 and 0.78 for classical machine learning multi-class classifiers. The baseline MLP model had an outstanding performance based on ROC-AUC and accuracy, with an accuracy of 0.79 and an ROC-AUC of 0.87. These results are in line with what has been proposed by [30,34,38], that deep learning models exhibit excellent detection accuracy and a low false-positive rate in the detection of attacks. However, noting that this

might be true, our focus is to develop an adversarial robust and explainable model such that users will have confidence in using these models.

**Table 4.** Baseline models performance.

| Method | Accuracy | F1 Score | AUC (Class = Normal) |
|---|---|---|---|
| Decision Tree Classifier | 0.71 | 0.74 | 0.79 |
| Random Forest Classifier | 0.71 | 0.72 | 0.78 |
| Linear SVM Classifier | 0.68 | 0.72 | 0.78 |
| Baseline MLP model | 0.79 | - | 0.87 |

*3.8. Adversarial Sample Generation*

Table 5 shows adversarial sample generation statistics. Three adversarial attacks, DeepFool, FGSM, and PGD, were employed to generate adversarial samples. Adversarial samples were generated from both train and test sets. An adversarial test set was used to test the resistance of baseline models to adversarial attacks. In contrast, a train set was used for adversarial training, a technique to develop an adversarial robust deep learning model.

**Table 5.** Statistics of adversarial test sample generation.

| Method | Number of Unique Features Changed | Number of Average Features Changed Per Datapoint | Average Perturbations/Sample (KDDTest+) |
|---|---|---|---|
| DeepFool | 122 | 51.73 | 0.10 |
| FGSM | 122 | 56.81 | 0.15 |
| PGD | 122 | 80.58 | 0.21 |

DeepFool parameters adopted in this research were: max iter: int = 100, epsilon: float = $1 \times 10^{-6}$, nb grads: int = 10, batch size: int = 1, verbose: bool = True, FGSM parameters were: estimator = classifier, eps = 0.6, mix ite = 100, targeted = False, batch size = 32, verbose = True and PGD parameters: estimator = classifier, eps = 0.6, max iter = 100, targeted = False, batch size = 32, verbose = True. On average, 56.58 features were changed per datapoint under FGSM attack, which was 5.08 more than the features change by DeepFool attack on average per data point. We obtained our dataset from [23]. Our research made use of: the full NSL-KDD train set including attack-type labels: *(KDDTrain+.TXT)*, which has 123,973 entries. The full NSL-KDD test set includes attack-type labels: *(KDDTest+.TXT)*, which has 22,544 entries. For our adversarial train set, *KDDTrain+* was used. The average perturbation was 0.1, 0.18, and 0.1 for DeepFool, PGD, and FGSM attacks, respectively. We used KDDTest+ to generate an adversarial test set. The average perturbations were 0.11, 0.21, and 0.15 for DeepFool, PGD, and FGSM attacks, respectively.

Feature Participation in Adversarial Sample Generation

Table 6 presents the top ten feature participations in adversarial sample generation. It can be noted that of the PGD top ten features, 40% ('*srv_count*', '*count*' '*dst_host_srv_count*', '*dst_host_diff_srv_rate*') also participated in the adversarial sample generation of DeepFool adversarial samples, while FGSM depicts a different pattern. Figure 8 presents a graphical representation of feature participation in PGD adversarial samples; *srv_count*' and '*count*' is shown to have had 100% participation in PGGadversarial sample generation.

**Table 6.** Feature participation top ten in adversarial example generation.

| Method | Features |
|---|---|
| DeepFool | 'srv_count', 'count', ' dst_host_srv_count', ' is_guest_login', 'dst_host_same_src_port_rate', 'root_shell', 'dst_host_diff_srv_rate', ' diff_srv_rate', 'dst_host_same_srv_rate', 'service _eco_i' |
| FGSM | 'land', 'dst_host_same_srv_rate', 'dst_host_count_ srv_rerror_rate', 'root_shell', 'dst_host_same_src_port_rate', 'protocol _icmp', 'dst_host_srv_serror_rate', 'service _ecr_i', 'num_outbound_cmds' |
| PGD | 'srv_count', 'count', 'dst_host_count', 'src_bytes', 'dst_bytes', 'dst_host_srv_count', 'dst_host_same_srv_rate', 'same_srv_rate', 'service _ssh', 'service _domain' |

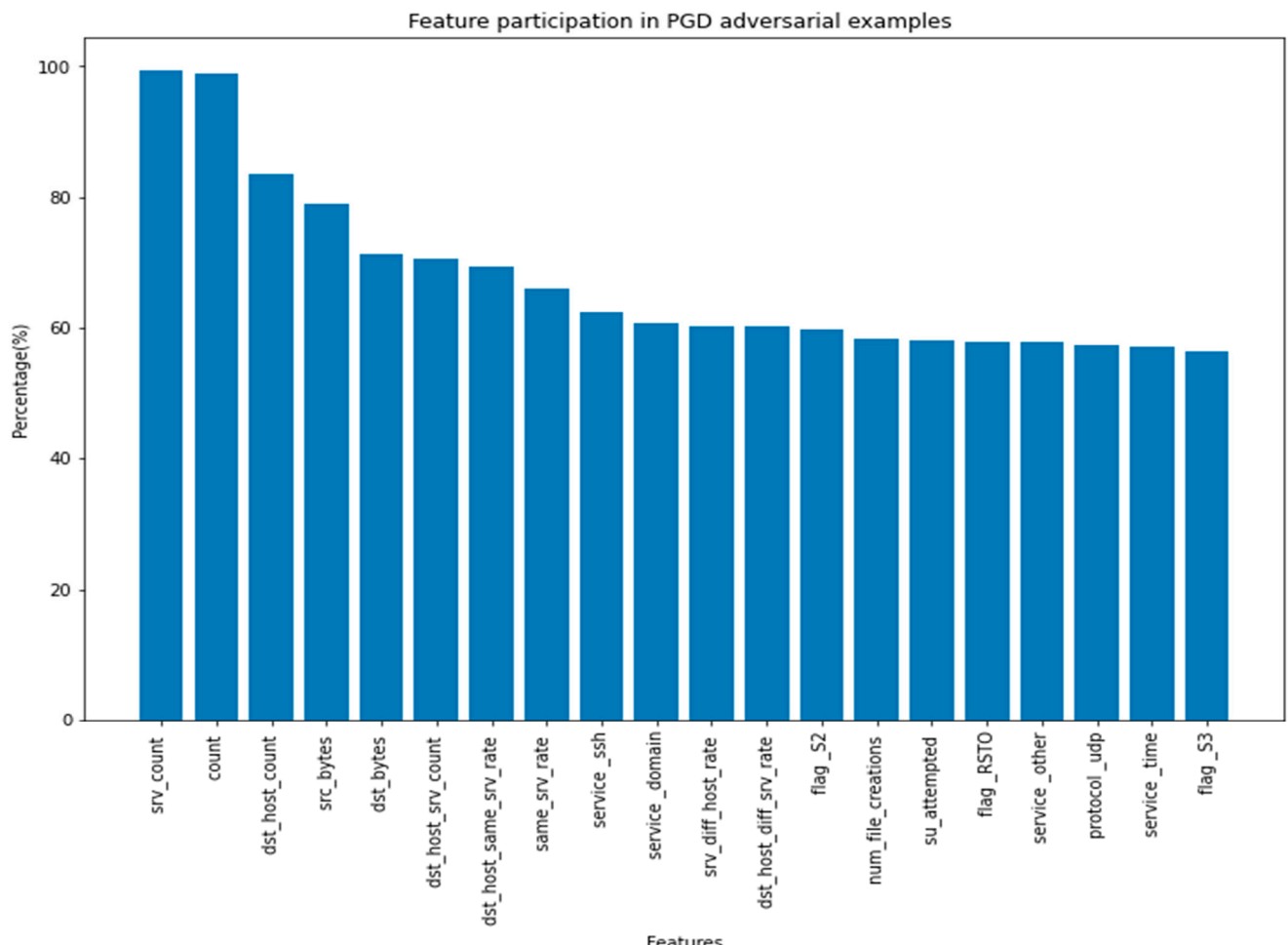

**Figure 8.** Feature participation in PGD adversarial examples.

### 3.9. Baseline Model Adversarial Resistant Evaluation

The adversarial robustness of baseline models is presented in Table 7. Four baseline models have been tested against three adversarial attacks: DeepFool, PGD, and FGSM. In Table 7, column, Normal, indicates ROC-AUC classification results using unaffected data, while DeepFool, FGAM, and PGD columns are ROC-AUC classification results under respective adversarial attacks. The general results indicate that our baseline models' performance is significantly affected by adversarial attacks, even for the baseline MPL model, which was used to generate the adversarial samples. There was a 31.03% reduction in the baseline MPL model's ROC-AUC under FGSM attack, with 29.89% and 28.74% under PGD and DeepFool, respectively. These results confirm that although deep learning models increase classification performance, they are vulnerable to adversarial attacks, which can undermine their performance and adoption. The results also showed that the PGD attack performed better on classical machine learning multi-class classifiers. It managed

to reduce the performance of the decision trees classifier by 65.82%, Random Forest by 70.51%, and Linear SVM by 70.51%, while DeepFool only managed to reduce classical machine learning multi-class classifiers by 17.71%, 32.05%, and 34.62%, respectively. FGSM was also inferior in attacking classical machine learning multi-class classifiers; its percentage reduction of decision trees classifier's ROC-AUC was 3.80% less than PGD and 3.85% less for both Random Forest and Linear SVM to PGD. These results align with [25], who confirmed that the FGSM attack's purpose is to be fast and to not have optimal attacks.

**Table 7.** Adversarial attacks evaluation on base models.

| Method | ROC-AUC | | | |
|---|---|---|---|---|
| | Normal | DeepFool | PGD | FGSM |
| Decision Tree | 0.79 | 0.65 | 0.27 | 0.30 |
| Random Forest | 0.78 | 0.53 | 0.23 | 0.26 |
| Linear SVM | 0.78 | 0.51 | 0.23 | 0.26 |
| Base MLP | 0.87 | 0.62 | 0.61 | 0.60 |

3.9.1. Adversarial Robustness Evaluation

Table 8 presents the major results of our research on the adversarial robustness of NIDS. We used our adversarial robust evaluation (10) to measure the robustness of adversarial-trained models.

**Table 8.** Adversarial attacks evaluation on adversarial-trained models.

| Method | ROC-AUC | | | | Adversarial Robust Evaluation | | | |
|---|---|---|---|---|---|---|---|---|
| | Normal | DeepFool | FGSM | PGD | Normal | DeepFool_A | FGSM_A | PGD_A |
| DeepFool_M | 0.86 | 0.86 | 0.84 | 0.73 | 0 | −0.20% | −2.33% | **−15.12%** |
| FGSM_M | 0.86 | 0.77 | 0.86 | 0.75 | 0 | −10.47% | 0.01% | **−12.79%** |
| PGD_M | 0.87 | 0.79 | 0.87 | 0.76 | 0 | **−9.20%** | **0** | **−13.65%** |
| Baseline MPL | 0.87 | 0.62 | 0.60 | 0.61 | 0 | −28.74% | −31.03% | −29.89% |

From Table 8, the figures under the block of columns below the ROC-AUC label, Normal, indicate ROC-AUC results for benign test samples, and DeepFool represents ROC-AUC results for DeepFool adversarial test samples. Likewise, FGSM represents the AUC results for FGSM adversarial test samples, and PGD represents ROC-AUC results for PGD adversarial test samples.

Columns that are below the Method label indicate the models under evaluation. For example, Normal indicates ROC-AUC adversarial-trained models, DeepFool_M indicates a DeepFool adversarial-trained model, FGSM_M indicates FGSM adversarial trained, and PGD_M indicates PGD adversarial-trained model.

The block of columns under the Adversarial Robust Evaluation label indicates the adversarial robust measure. This evaluation follows Equation (12) in percentage terms. It measures the rate of change of a model's ROC-AUC after the model has been attacked by an adversarial attack. The Normal column indicates no attack, meaning that the classification or prediction process is being performed using benign samples. The DeepFool_A column indicates the rate of change of the respective model's ROC-AUC after the DeepFool attack. The FGSM_A column indicates the rate of change of the respective model's ROC-AUC after the FGSM, and PGS_A indicates the rate of change of the respective model's ROC-AUC after the PGD attack.

The overall results indicate that the adversarial-trained models are more robust to new adversarial samples and outperform the baseline MPL. This confirms the work of

[2,3,19,25,39]. From Table 8, it can be noted that all adversarial attacks greatly reduced baseline MPL's ROC-AUC; under PGD attack, it was reduced by 29.89%, while under FGSM, it was reduced by 31.03%, and under DeepFool attack, it was reduced by 28.74%.

PGD adversarial-trained model was a more robust mode than DeepFool- and FGSM-trained models. FGSM attack did not impact the PGD-trained model; there was no change to the PGD-trained model's ROC-AUC after the FGSM attack. The DeepFool attack caused minimal damage to a PGD adversarial-trained model with a −9.20% reduction to the PGD adversarial-trained model's ROC-AUC compared to what the PGD adversarial attack caused to the DeepFool adversarial-trained model. PGD attack caused a 15.12% reduction in the DeepFool adversarial-trained model and a 12.79% reduction in the FGSM-trained model. PGD attack also caused a 13.65% reduction in the PGD adversarial-trained model's ROC-AUC; this indicates that PGD adversarial attack has higher strength because it managed to reduce PGD adversarial-trained model's ROC-AUC more than DeepFool and FGSM.

The Table 8 results can also be used to measure the strength of the adversarial attacks under review; it can be noted from Table 8 that FGSM attacks have the least strength when attacking adversarial-trained models. However, it was unable to affect the performance of the PGD adversarial-trained model, as there was no change in the ROC-AUC. FGSM also managed a mere 2.33% reduction in DeepFool adversarial-trained model's ROC-AUC. A further illustration of the adversarial-trained model's adversarial robustness is presented in Appendix A, Figures A1–A5. The figures show AUC results for the adversarial-trained models for all the classes (normal, DOS, R2L Probe, and U2R). The results indicate similar patterns as presented in Table 8, where ROC-AUC was calculated as a micro average of all five classes. These results infer that the PGD adversarial-trained model is more robust than the DeepFool and FGSM adversarial-trained model; hence, our final robust model is the PGD adversarial-trained model.

### 3.9.2. Robust Model Local and Global Explanation Results

After choosing the adversarial robust model, the final stage was to present explanations about the model—local and global explanations. We used the SHapley Additive exPlanations (SHAP) framework to perform this task.

Figure 9 presents the robust model local explanation results. We chose *prob* attacks as an example to detect how the model classified an attack as *prob*. Five hundred samples were randomly selected from our dataset, and each feature's average Shapley values were calculated. The contribution results are presented in Figure 9. It is shown that when the model is 56% sure that the attack is *prob*, *dst_byte*, *src_bytes*, *dst_host_srv_count*, *and count*, it would have contributed immensely to the decision.

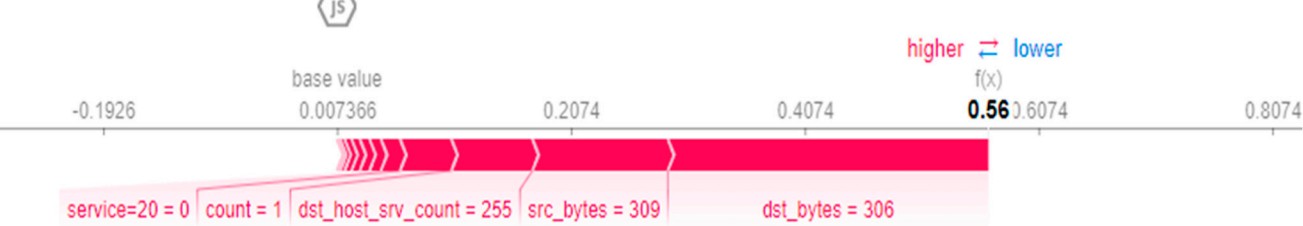

**Figure 9.** Interpretation of the adversarial robust model on Prob attack.

### 3.9.3. Robust Model Global Explanation Results

This section details the global explanation results of our adversarial robust model, presented in Appendix B and Appendix C. The summary plots represent the Shapley value for a feature and an instance. Features determine the position on the *y*-axis, while the Shapley value determines the *x*-axis. Features in the *y*-axis are arranged according to

their importance [30]. The most important feature is found at the top of the summary plot, while the least important one is found at the bottom. The colors represent the magnitude of the feature value. The red color indicates a higher value; as the red color intensifies, the value of the feature also increases, whereas the blue color signifies the least valued features, as the intensity of the blue color increases, the value of the feature also decreases.

In Appendix B, Figure A6a shows a summary plot of the top 20 features extracted for DOS, and Figure A6b shows a summary of the top 20 features extracted for R2L. While in Appendix C, Figure A7a shows a summary of the top 20 features extracted for prob, and Figure A7b shows the top 20 features extracted for U2R. The summary is presented in Table 9.

**Table 9.** A summary of the four attack types extracted by the adversarial robust NIDS.

| Attack Label | Important Features Extracted by the Adversarial Robust NIDS |
|---|---|
| Denial of service (DOS) | dst-bytes, src_byte, count, dst_hiost_srv_count, dst_host_count, duration, srv_count, flag = 5, diff_srv_rate, dst_host_diff_srv_rate, dst_host_same_src_port_rate, dst_host_src_serror_rate, dst_host_serror_rate, srv_serror_rate, flag = 9, serror_rate, wrong_fragment, same_srv_rate, service = 49, flag = 9 |
| Prob | dst_bytes, src_bytes, count, dst_host_srv_count, dst_host_count, duration, dst_host_same_src_port_rate, dst_host_diff_srv_rate, srv_count, diff_srv_rate, flag = 5, flag = 9, dst_host_srv_seror_rate, protocol_type = 0, dst_host_serror_rate, service = 49, serror_rate, srv_serro_rate, service = 14, same_srv_rate |
| Remote to local (R2L) | src_bytes, dst_bytes, dst_host_srv_count., dst_host_count, duration, count, srv_count, flag = 5, num_compromised, dst_host_srv_diff_host_rate, service = 66, dst_host_same_rv_rate, service = 44, flag = 1, service = 20, service = 24, service- = 65, service- = 51, dst_host_rerror_rate |
| User To Root (U2R) | src_bytes,dst_bytes, dst_host_srv_count, dst_host_count, count, duration, srv_count,flag = 5, dst_host_srv_diff_host_rate, service = 44, service = 18, service = 24, flag = 9, dst_host_same_srv_rate, diff_srv_rate, service = 51, logged_in, wrong_fragment, srv_serror_rate, protocol_type = 0 |

## 4. Discussion

This research work used the NSL-KDD dataset to develop a robust and explainable NIDS based on deep learning. Three untargeted and white-box attacks were used to generate adversarial examples for the experiment. We demonstrated how adversarial attacks could undermine classical multi-class machine learning-based NIDS as well as deep learning-based NIDS, and this confirms the works of [7–10,20]. Hence, we find justification for creating an adversarial robust deep learning model that is less affected by adversarial attacks.

Our adversarial robust and explainable NIDS based on deep learning was developed under the white-box scenario with only three adversarial attacks *(FGSM, PGD, and Deep-Fool)*. In addition, we used the NSL-KDD dataset because of its size and considerable number of attacks; however, it does not contain modern attack types. Hence, our current model suffers from generalizability. However, this can be solved by retaining the model with modern network security datasets under a large pool of adversarial attacks. Furthermore, we used an epsilon of 0.6 as one of the adversarial attacks hyperparameters; we adopted Debicha et al. (2021) [19], who indicated that increasing epsilon tends to be insignificant in terms of reducing the classifier's accuracy. However, this might not guarantee the stability of adversarial attacks; we propose further research to focus on hyperparameter tuning on all the adversarial attacks to investigate the stability of an adversarial robust model.

While the results of previous experiments have shown that adversarial training increases the robustness of network intrusion detection systems [19,28], we observed that few researchers have focused on robust adversarial training, that is, testing the resistance

of adversarial-trained models with adversarial samples from different adversarial attacks. This paper proposes a method to measure adversarial robustness; Equation (12). Our main metric is the ROC-AUC. We also followed standard classification metrics proposed by [31]. We acknowledge that there should be robust classification measures for cybersecurity models to substantiate the reliability of the classification results. Using method (12), we tested the adversarial robustness of the FGSM adversarial-trained model, DeepFool adversarial-trained model, and PGD adversarial-trained model. We observed that the PGD adversarial-trained model is more robust than the DeepFool adversarial-trained model and FGSM adversarial-trained model. These results confirm [3], who indicated that a model trained to be robust against PGD adversaries will be robust against a wide range of attacks. Furthermore, our results also confirmed [40], who developed a state-of-the-art defense model based on a PGD adversarial-trained model as the backbone model.

## 5. Conclusions

Creating adversarial, robust, and explainable DNN-based NIDS is a major step in ensuring a safe digital environment. The research work proposes a novel measure of adversarial robustness, Equation (12), for DNN adversarial robustness comparison. We also propose an adversarial robust and explainable network intrusion detection system based on deep neural networks by implementing explainable AI techniques and adversarial machine learning into NIDS. The overall results indicate that the adversarial-trained models are more robust to new adversarial samples and outperform the baseline MPL; this confirms the work of [2,3,19,25,39]. PGD adversarial-trained model was a more robust model than DeepFool- and FGSM-trained models. We implemented the SHAP technique to explain adversarial robust NIDS based on DL to extract important features used by the model to make classification decisions. We strongly believe that this is the first paper to implement a combination of explainable AI techniques and adversarial learning into IDS. Our adversarial robust and explainable NIDS based on deep learning was developed with a minimum number of adversarial attacks. Under the white-box scenario, we also used the NSL-KDD dataset because of its size and considerable number of attacks; however, it does not contain modern attacks. Hence, our current model suffers from generalizability. This predicament can be solved by retaining the model with modern network security datasets under a large pool of adversarial attacks. Future works should incorporate more novel network traffic datasets with more attacks to obtain a good measure of the impact of adversarial sample generation, thereby working toward a model that can be generalized.

**Author Contributions:** Conceptualization, K.S.; methodology, K.S.; software, K.S.; validation, G.-Y.S., D.-W.K. and M.-M.H.; formal analysis, K.S.; investigation, K.S.; resources, G.-Y.S. and M.-M.H.; data curation, K.S.; writing—original draft preparation, K.S.; writing—review and editing, K.S., G.-Y.S.., D.-W.K., and M.-M.H.; visualization, K.S.; supervision, M.-M.H.; project administration, G.-Y.S. and M.-M.H.; funding acquisition, G.-Y.S., D.-W.K. and M.-M.H. All authors have read and agreed to the published version of the manuscript.

**Funding:** This research was supported by National Research Foundation of Korea (NRF) grant funded by the Korea government (MSIT) (No. 2022R1F1A1073375).

**Institutional Review Board Statement:** Not applicable.

**Informed Consent Statement:** Not applicable.

**Data Availability Statement:** The data presented in this study are openly available in Canadian Institute for Cybersecurity. This dataset cabn be found here https://www.unb.ca/cic/datasets/nsl.html (accessed on 28 April 2022).

**Acknowledgments:** We would like to appreciate UCI Knowledge Discovery in Databases Archive (https://kdd.ics.uci.edu/) for their effors for providing clear explanation of KDD99 dataset.

**Conflicts of Interest:** The authors declare no conflict of interest.

**Appendix A**

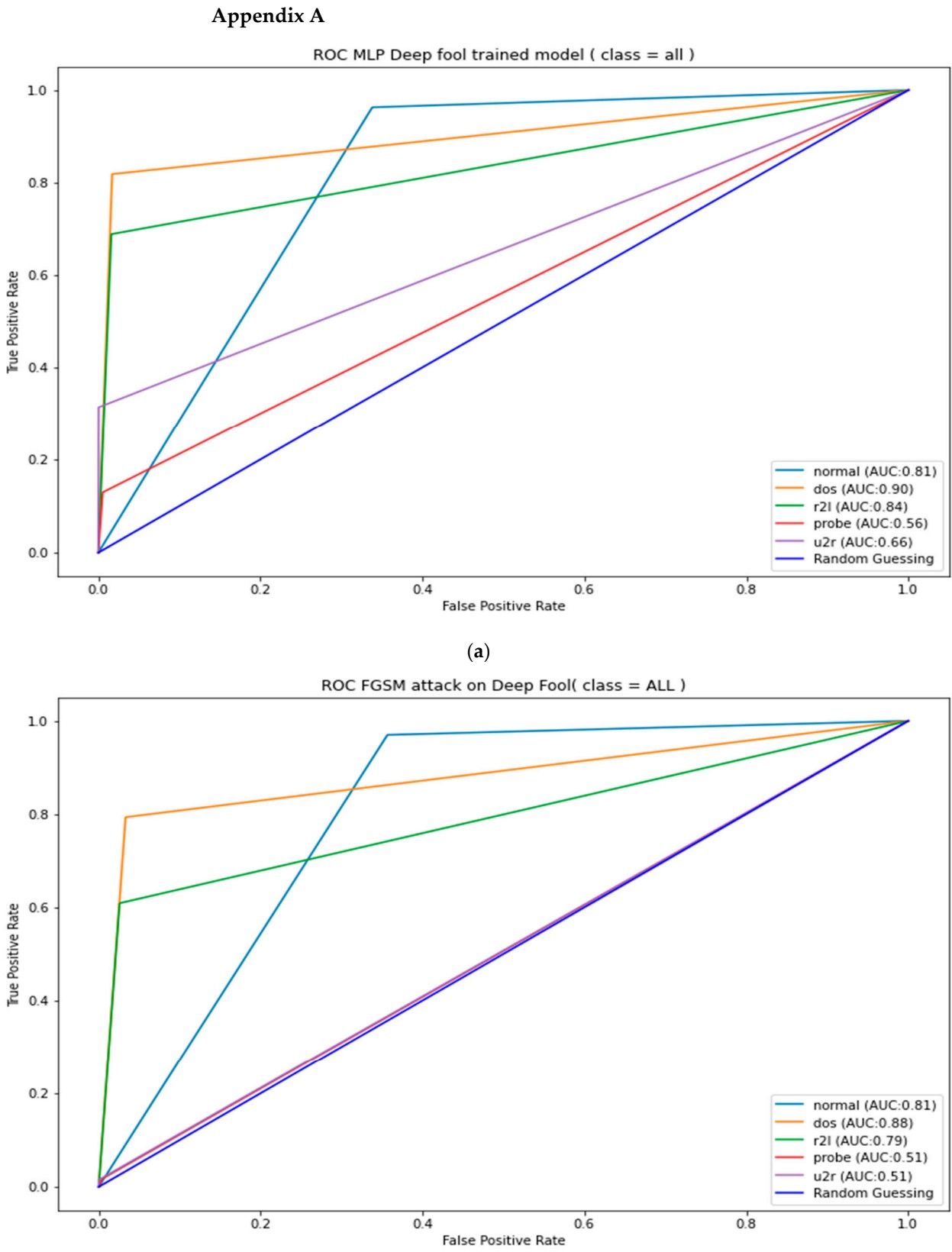

(**a**)

(**b**)

**Figure A1.** DeepFool adversarial-trained model ROC-AUC (**a**). DeepFool robustness test against FGSM (**b**).

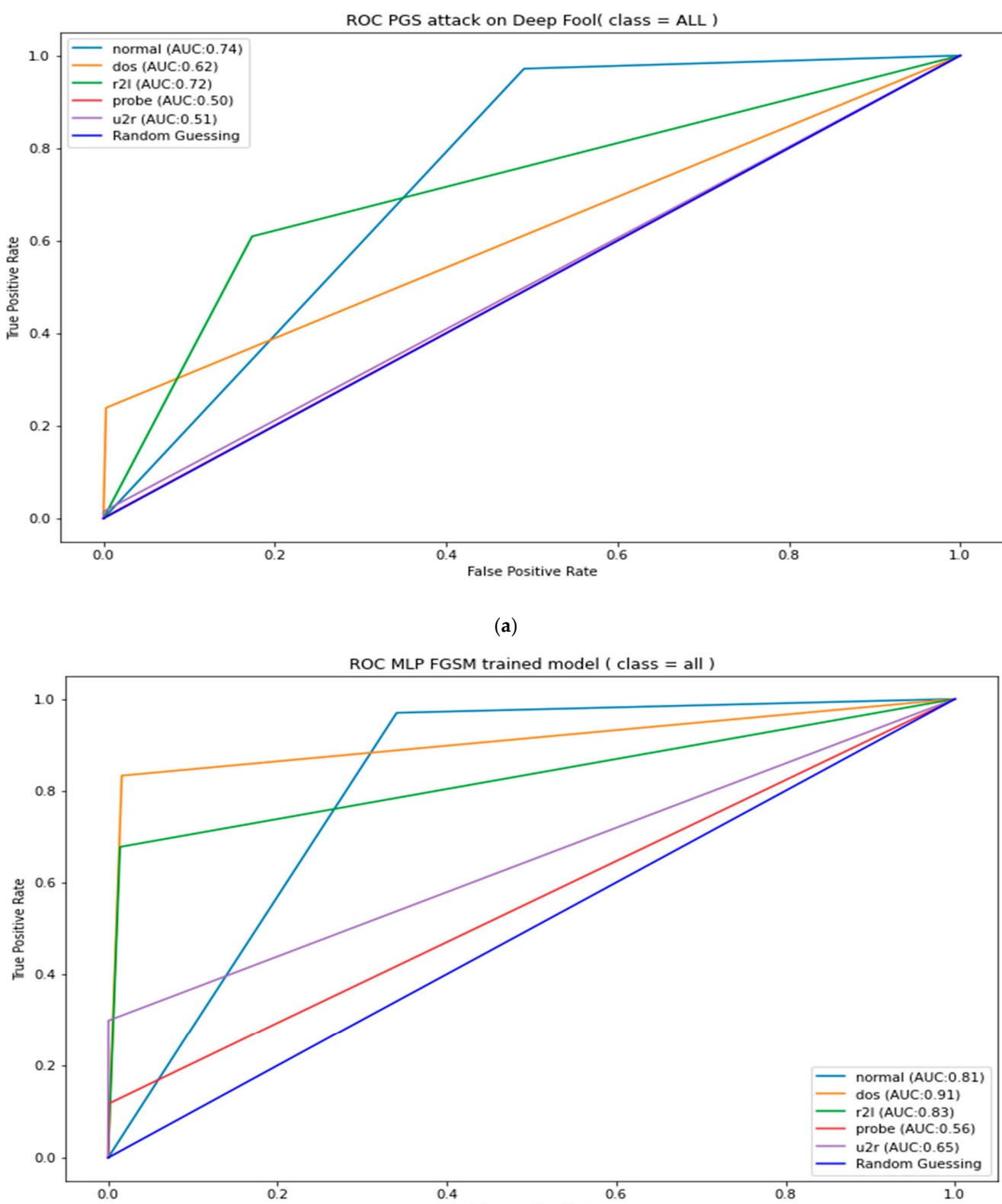

(**a**)

(**b**)

**Figure A2.** DeepFool robustness test against PGD attacks (**a**). FGSM adversarial-trained model ROC-AUC (**b**).

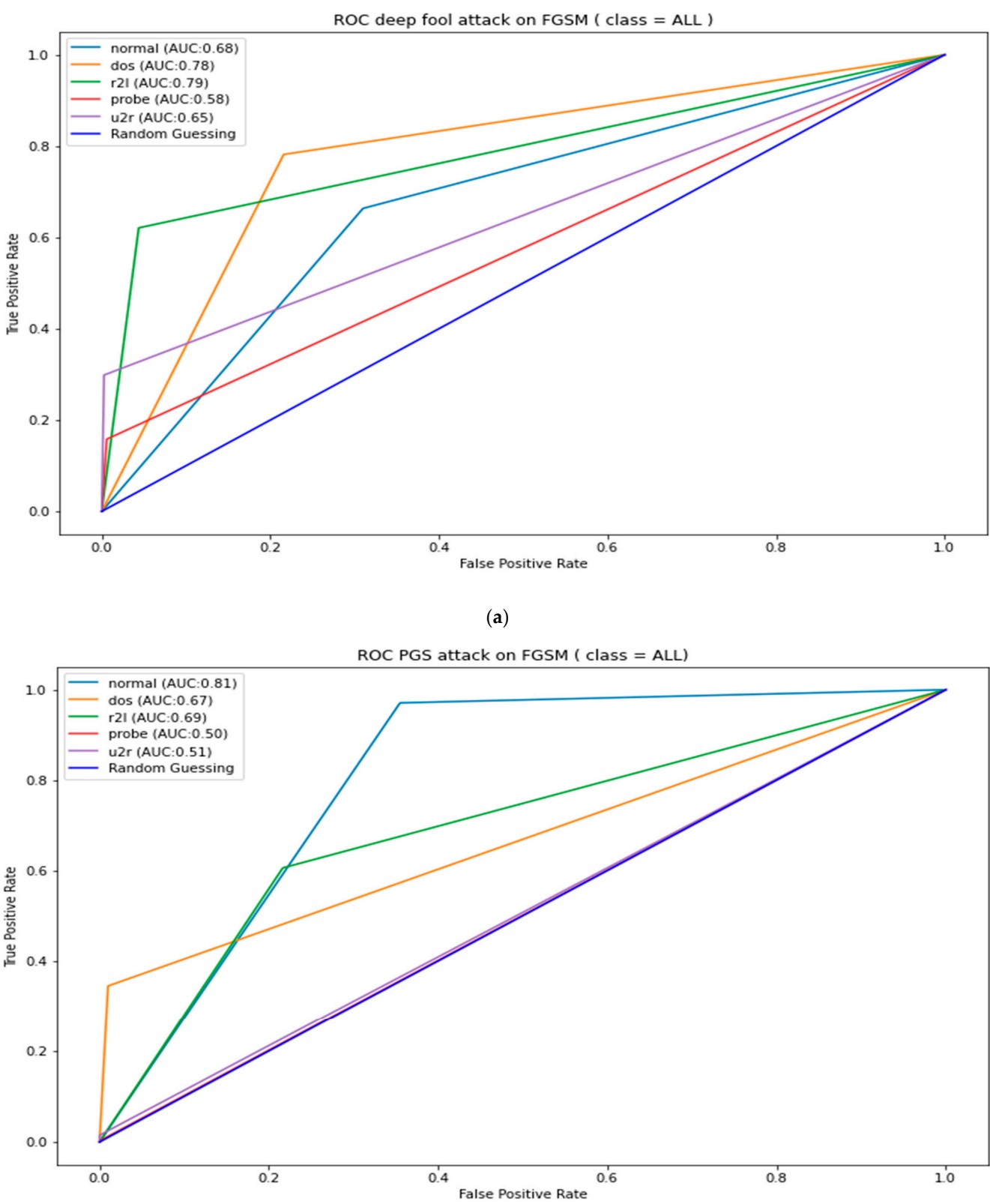

**Figure A3.** FGSM robustness test against DeepFool (**a**) and PGD attacks (**b**).

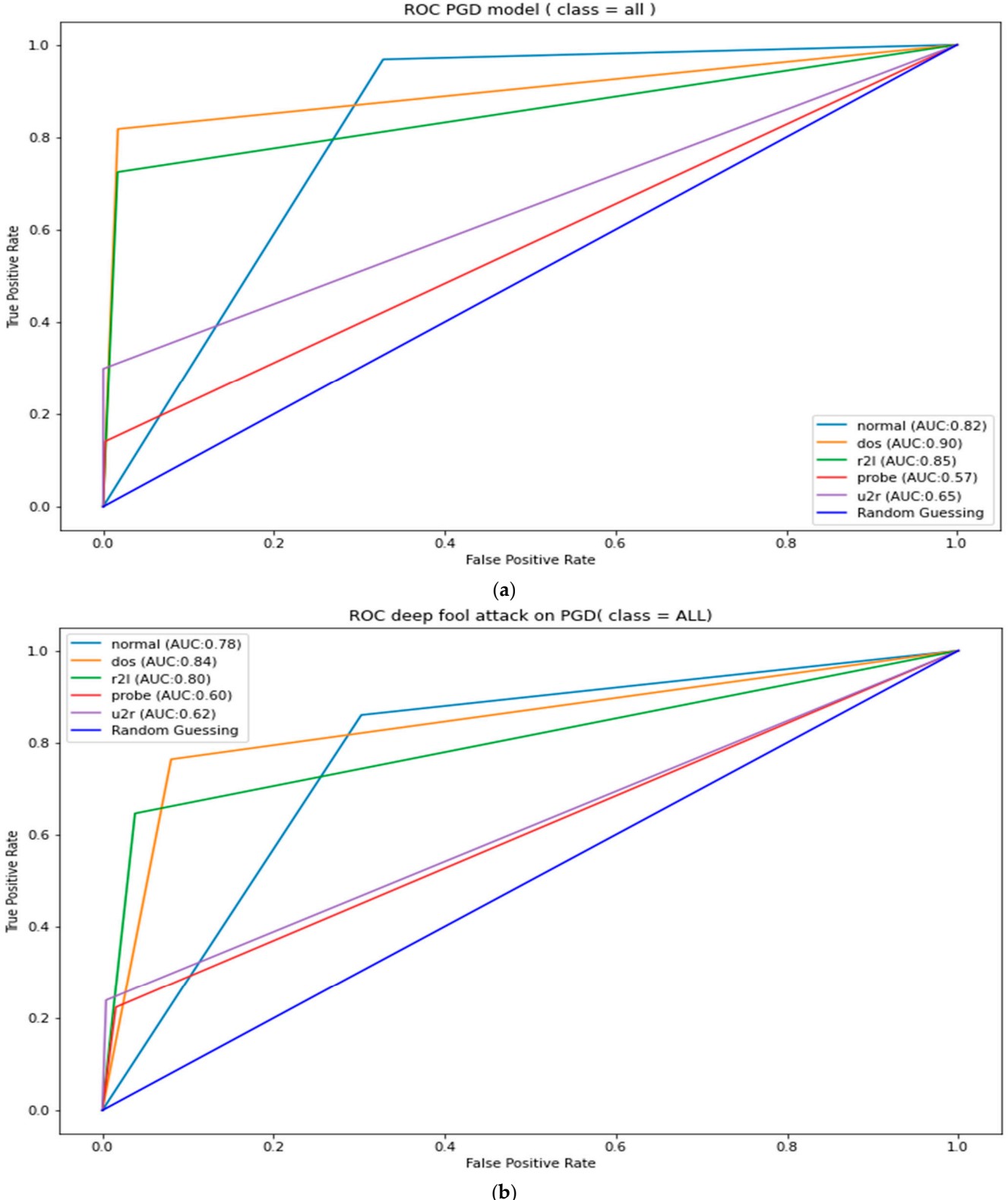

**Figure A4.** PGD adversarial-trained model ROC-AUC (**a**). PGD robustness test against DeepFool attacks (**b**).

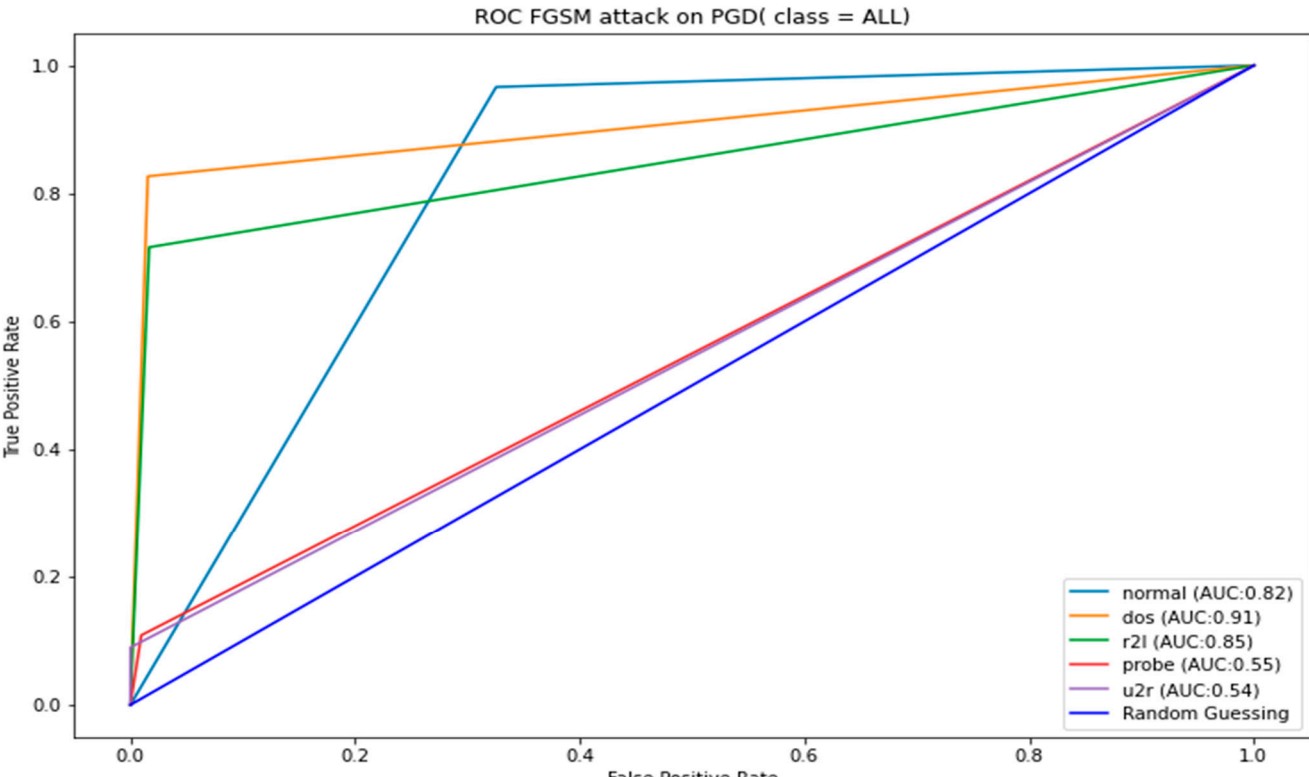

**Figure A5.** PGD robustness test against FGSM attacks.

## Appendix B

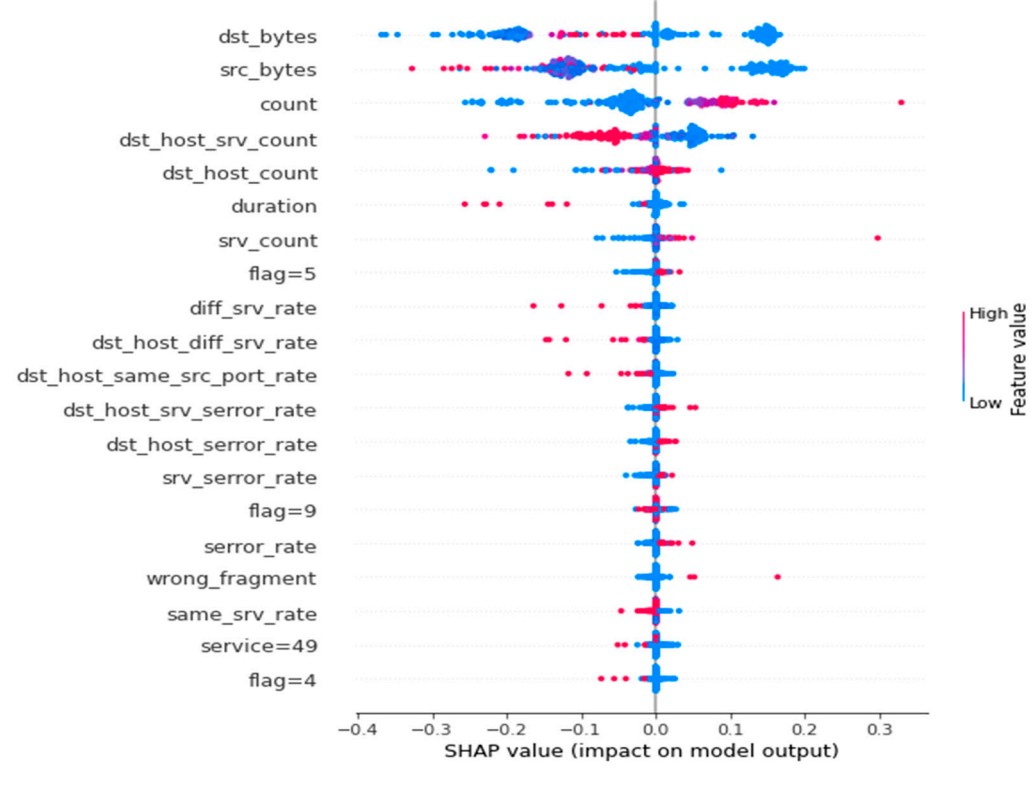

(**a**) DOS

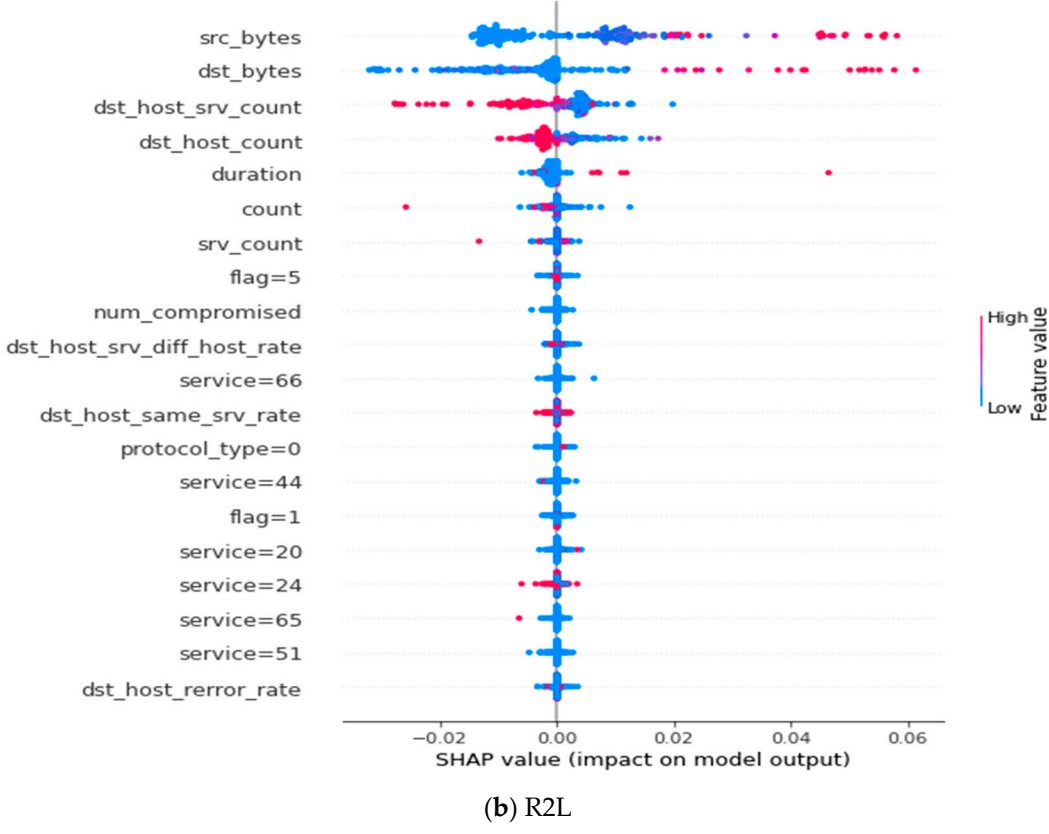

(**b**) R2L

**Figure A6.** Top 20 important features of (**a**) DOS and (**b**) R2L.

**Appendix C**

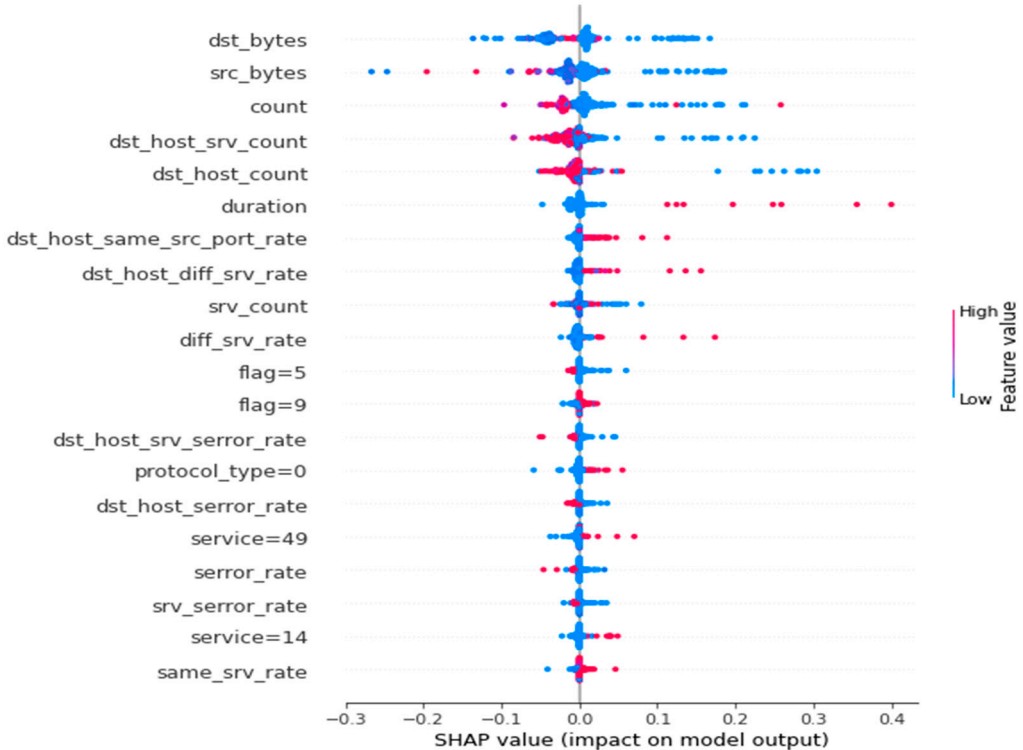

(**a**) prob.

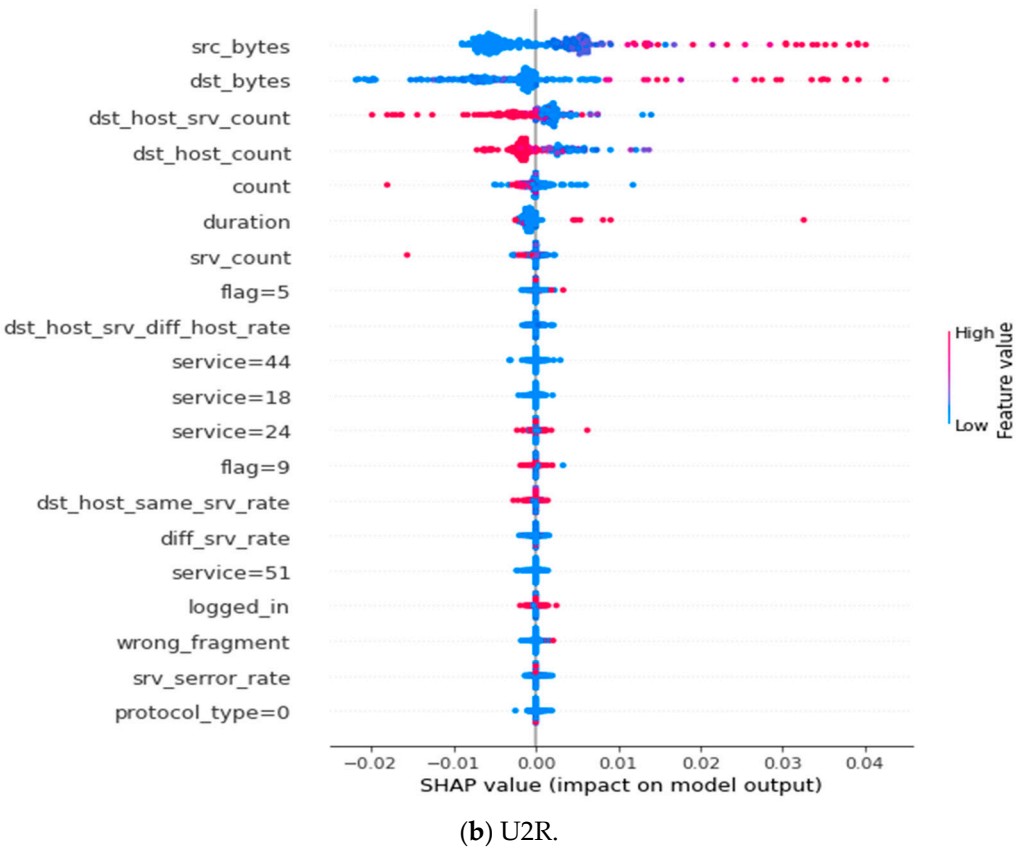

(**b**) U2R.

**Figure A7.** Top 20 important features of (**a**) prob and (**b**) U2R.

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
