# Peer review of "Adversarial Robust and Explainable Network Intrusion Detection Systems Based on Deep Learning"

_applsci, doi:10.3390/app12136451_

Round 1

Reviewer 1 Report

The paper investigates the robustness of NIDS systems to adversarial attacks. The authors focus their experiments on adversarial training and its results to robustness, which can be interesting to the community, but there are many other approaches in practice to increase robustness.

The main goal of the paper is understandable, but unfortunately the paper is badly structured and written. It contains a fairly high number of grammatical errors or vague expressions. Most of these are minor, but here are some which hinders the understanding of some sentences (or at least or me). Expressions like:
"or compromise of malicious when modifying malicious traffic" malicious what?
"extends his work to a more adversarial attack method" How is a method more (or less) adversarial than an other?

"and redundant are dropped." are redundant elements dropped? or is redundancy lowered?

"The attacks at the prediction phase are called poisoning attacks, while those which are crafted at the prediction phase are called evasion
attacks " The attacks at the training phase are called poisoning so this is factually wrong. The same thing is explained again (correctly, but unnecessarily) in the next paragraph.

These are just some of the examples and because of this the structure and the grammar of the paper has to be improved.

Also there are some structural problems with the manuscript. The results are summarized in the introduction, which is not a  common thing (and I would not do that usually they are summarized in the abstract and the conclusion), but this is acceptable.

It is strange for me that the authors list some recent and well performing adversarial attacks (such as C&W or JSMA) but they do not describe or investigate them.

It would be good if the citation of the other, investigated methods (FGSM, PGD) would be done at their first appearance (at the top of page 2).

Apart from the structural problems unfortunately there are some theoretical problems as well, which has to be corrected:

The performance of the investigated attack methods depend heavily on their parametrization.  It was unclear for me what were the exact parameters of the investigated method (this is important for the sake of reproducibility) and how their performance and the robustness of the system depends on these parameters.

Also the creation of the trainingset is unclear. How many samples were used in the dataset? Were all samples successfully attacked by all attack mechanism or how where they selected? What was the success rate of these attack on the original samples?

Also in table five the generation time of the whole dataset is not relevant, it would be good to know the dataset size and the execution time should be reported per sample.

Why are the same attack mechanisms not reported in table 8. Normally in adversarial training some attack samples should be used in training to increase robustness and validation should happen on independent samples. If independent samples are used the effect of the same mechanisms can also be measured (and usually it is a useful reference point, since he robustness will not be 100% in this case either).

It would also be good to investigate how robust these results are. Will they remain consistent if I change the epsilon parameter or the iteration number in FGSM? I am afraid not. This should be investigated how consistent the results are and how much they depend on the algorithmic parameters.

Unfortunately without the modification and addition of these points I can not wholeheartedly support the acceptance of the paper.

Reviewer 2 Report

The article is devoted to the application of machine learning methods for solving cybersecurity issues. The topic of the article is relevant. The structure of the article is classical. All the necessary sections are present in the article. The level of English is acceptable. The article is easy to read. The quality of the figures is poor. The text on them is blurry. The article cites 41 relevant sources.

The following recommendations can be formulated for the material of the article:

  1. Of the 41 sources cited, 23 are mentioned in the introduction. At the same time, sections 2.1-2.4 are overviews. Their place is in the introduction or in the state-of-art section.
  2. In the well-known article "Artificial Neural Network for Cybersecurity: A Comprehensive Review", an extensive study was carried out to identify threats using neural networks. The review article considered the results of models for classifying attacks on datasets with network connection data, such as KDD Cup 99, NSL-KDD, Alexa, OSINT, etc. The best results were shown by LSTM, CNN-based architecture models, BiLSTM and Autoencoder. Therefore, this article proves the concept of successfully using neural networks to detect threats with a fairly high accuracy. Judging by the architecture of the author's classifier (Fig. 6), the authors obtained different results. I ask you to technically correctly justify the differences.
  3. To determine the 4 types of attacks, the authors analyze 41 characteristic parameters. This is clearly redundant. How do the authors ensure the robustness of the classification? The study clearly lacks the use of factor analysis. One of its progressive variants (applicable for the author's neural network architecture) is the use of a “bottle-neck” layer before the penultimate internal classifier layer.
  4. I really liked that the authors did not forget about ROC curves. At the same time, the authors cited about fifteen graphs with the results of experiments, and the size of the Discussion section is less than half a page. Either a lot of unnecessary experiments have been carried out, or the authors are hiding something ) Please resolve this paradox.
  5. The authors used the metric (6)-(10) typical for assessing the quality of neural network classification. At the same time, cybersecurity is a little more serious than distinguishing dogs from cats ) It is necessary to substantiate the reliability of the results obtained from the standpoint of the provisions of mathematical statistics using the appropriate criteria.

Round 2

Reviewer 1 Report

I would like to thank the authors for considering my comments and implementing them. I think they have significantly increased the quality of the manuscript.

my opinion is that content-wise the manuscript can be accepted. I would be glad to see how the results depend on the parameters from the attacks, and how general the results are but this could be investigated in an other paper. The paper in its current form has added value and can be interesting for people working in the field of intrusion detection systems.

Unfortunately there are still some grammatical errors in the manuscript, such as:

"Iinput layer is depicted" or "main evaluation matric" . These errors has to be corrected in the final version.

Reviewer 2 Report

I had the following recommendations for the previous version of the article:

  1. In the well-known article "Artificial Neural Network for Cybersecurity: A Comprehensive Review", an extensive study was carried out to identify threats using neural networks. The review article considered the results of models for classifying attacks on datasets with network connection data, such as KDD Cup 99, NSL-KDD, Alexa, OSINT, etc. The best results were shown by LSTM, CNN-based architecture models, BiLSTM and Autoencoder. Therefore, this article proves the concept of successfully using neural networks to detect threats with a fairly high accuracy. Judging by the architecture of the author's classifier (Fig. 6), the authors obtained different results. I ask you to technically correctly justify the differences.
  2. To determine the 4 types of attacks, the authors analyze 41 characteristic parameters. This is clearly redundant. How do the authors ensure the robustness of the classification? The study clearly lacks the use of factor analysis. One of its progressive variants (applicable for the author's neural network architecture) is the use of a “bottle-neck” layer before the penultimate internal classifier layer.
  3. I really liked that the authors did not forget about ROC curves. At the same time, the authors cited about fifteen graphs with the results of experiments, and the size of the Discussion section is less than half a page. Either a lot of unnecessary experiments have been carried out, or the authors are hiding something) Please resolve this paradox.

4: The authors used the metric (6)-(10) typical for assessing the quality of neural network classification. At the same time, cybersecurity is a little more serious than distinguishing dogs from cats ) It is necessary to substantiate the reliability of the results obtained from the standpoint of the provisions of mathematical statistics using the appropriate criteria.

I note that the authors generally took them into account well in the updated version of the work.

Of course, the article can still be improved on formal grounds (for example, clearly formulate the scientific novelty in the conclusions), but I believe that the article can already be published.
